# On Causal Discovery in the Presence of Deterministic Relations

**Loka Li**[1*], **Haoyue Dai**[2*], **Hanin Al Ghothani**[1], **Biwei Huang**[3],
**Jiji Zhang**[4], **Shahar Harel**[5], **Isaac Bentwich**[5], **Guangyi Chen**[1,2], **Kun Zhang**[1,2]
[1] Mohamed bin Zayed University of Artificial Intelligence
[2] Carnegie Mellon University, [3] University of California San Diego
[4] The Chinese University of Hong Kong, [5] Quris AI
{longkang.li, kun.zhang}@mbzuai.ac.ae

## Abstract

Many causal discovery methods typically rely on the assumption of independent noise, yet real-life situations often involve deterministic relationships. In these cases, observed variables are represented as deterministic functions of their parental variables without noise. When determinism is present, constraint-based methods encounter challenges due to the violation of the faithfulness assumption. In this paper, we find, supported by both theoretical analysis and empirical evidence, that score-based methods with exact search can naturally address the issues of deterministic relations under rather mild assumptions. Nonetheless, exact score-based methods can be computationally expensive. To enhance the efficiency and scalability, we develop a novel framework for causal discovery that can detect and handle deterministic relations, called Determinism-aware Greedy Equivalent Search (DGES). DGES comprises three phases: (1) identify minimal deterministic clusters (i.e., a minimal set of variables with deterministic relationships), (2) run modified Greedy Equivalent Search (GES) to obtain an initial graph, and (3) perform exact search exclusively on the deterministic cluster and its neighbors. The proposed DGES accommodates both linear and nonlinear causal relationships, as well as both continuous and discrete data types. Furthermore, we investigate the identifiability conditions of DGES. We conducted extensive experiments on both simulated and real-world datasets to show the efficacy of our proposed method. The code is available at `https://github.com/lokali/DGES.git`.

## 1 Introduction

Causal discovery from observational data has attracted considerable attention in recent decades and has been widely applied in various fields such as machine learning [1], healthcare [2], manufacturing [3] and neuroscience [4]. Most causal discovery methods operate under the assumption of independent noises in the probabilistic system. However, real-world scenarios frequently encounter deterministic relationships. For example, the body mass index (BMI) is defined as the weight divided by the square of the body height, composing a deterministic relation among weight, height, and BMI.

Constraint-based and score-based methods are two primary categories in causal discovery. Constraint-based methods, such as PC [5] and FCI [6], leverage conditional independence tests (CIT) to estimate the graph skeleton and then determine the orientation. Under the Markov and faithfulness assumptions [7], these methods are guaranteed to asymptotically output the true Markov equivalence class (MEC). However, the faithfulness assumption is sensitive to many factors, such as the statistical errors with finite samples. Moreover, in the presence of deterministic relations, the faithfulness assumption is

---

*Equal contributions.

38th Conference on Neural Information Processing Systems (NeurIPS 2024).

always violated. Take the chain structure $X \to Y \to Z$ for example where $Y = f(X)$. In this case, faithfulness is violated due to the conditional independence $Z \perp\!\!\!\perp Y|X$, i.e., when $X$ is given, $Y$ degenerates to a constant that is independent to any variables. Several variants of constraint-based methods [8, 9] have been proposed to accommodate certain types of unfaithfulness. However, they generally provide practical flexibility but do not guarantee the identification to the true MEC.

For score-based methods, the approach can vary based on the search strategy, which may involve greedy search, exact search, or continuous optimization. One typical score-based method with greedy search is Greedy Equivalent Search (GES) [10], which searches in the space of MECs greedily by maximizing a well-defined score, such as Bayesian information criterion (BIC) score [11]. Specifically, GES starts with an empty graph and consists of two phases. In the forward phase, it incrementally adds one edge at a time if it yields the maximum score improvement, continuing until no further edge can be added to enhance the score. In the backward phase, it checks all edges to eliminate some if removal further improves the score. Similar to the aforementioned constraint-based methods, GES converges to the true MEC in the large sample limit.

Some exact score-based methods aim at weakening the faithfulness assumption required for asymptotic correctness of the search results, such as dynamic programming (DP) [12, 13], A* [14, 15], and integer programming [16, 17]. The DAGs estimated by these methods can be converted to their MECs for causal interpretation [18]. Lu et al. [19] demonstrated that these exact methods may produce correct results in cases where methods relying on faithfulness fail. Furthermore, Ng et al. [20] proved that exact score-based search with BIC can asymptotically outputs the true MEC when the sparsest Markov representation (SMR) assumption [21] is satisfied. Note that the SMR assumption is strictly weaker than the faithfulness assumption.

Deterministic relations have been considered in a few works of causal discovery. D-separation condition [7] is proposed for graphically determining conditional independence. Glymour [22] proposed a heuristic procedure to learn the causal graph in a deterministic system, called DPC, where only a subset of variables will be conditioned in testing conditional independence. Daniusis et al. [23] and Janzing et al. [24] considered a deterministic system with only two variables, and presented the idea of independent changes to infer the causal direction. Luo [25] and [26] incorporated the classical PC algorithm and utilized additional independence tests to handle determinism. Mabrouk et al. [27] combined a constraint-based approach with a greedy search that included specific rules to deterministic nodes and significantly reduce the incorrect learning. However, there is no identifiability guarantee in those related works. Moreover, Zeng et al. [28] assumes nonlinear additive noise model under high-dimensional deterministic data while Yang et al. [29] assumes linear non-Gaussian model. Different from them, this paper aims to provide a principled framework to handle deterministic relations for arbitrary functional models. More related works are given in Appendix A2.

**Contributions.** Firstly, we find that exact score-based methods can naturally be used to address the issues of deterministic relations when mild assumptions are fulfilled. Secondly, due to the large search space of the possible DAGs, the exact score-based methods are feasible only for small graphs and can be inefficient for large graphs. To enhance the efficiency and scalability, we propose a novel framework called **D**eterminism-aware **G**reedy **E**quivalent **S**earch (DGES), aimed at enhancing the efficiency and scalability to handle deterministic relations. Importantly, DGES is a general three-phase method, with no restricted assumption on the underlying functional causal models, i.e., it can accommodate both linear and nonlinear relationships, Gaussian and non-Gaussian data distributions, as well as continuous and discrete data types. Thirdly, we provide the identifiability conditions of DGES under general functional models. Last but not least, we conducted extensive experiments on both simulated and real-world datasets to validate our theoretical findings and show the efficacy of our proposed method.

**Paper organization.** In Section 2, we review the common assumptions, provide a motivating example why PC fails in dealing with deterministic relations, then present our intuitive solution using exact score-based method. In Section 3, we present our proposed DGES with three phases in details. Furthermore, we provide the identifiability conditions for DGES presented in a general form in Section 4. The empirical studies in Section 5 validate our theoretical results and show the efficacy of our method. Finally, we conclude our work with further discussions in Section 6.

## 2 Causal Discovery with Deterministic Relations

In this section, we first review the preliminaries of causal discovery, especially with deterministic relations, and then we provide some common assumptions that are related to our further analysis, as presented in section 2.1. Furthermore, we display two scenarios with deterministic relations where faithfulness can be violated in section 2.2, explaining why using constraint-based methods such as the PC algorithm can be problematic in addressing deterministic issues. Lastly, we provide an intuitive solution to handle the deterministic issues by exact score-based methods, as shown in section 2.3.

### 2.1 Causal Discovery and Common Assumptions

Let $\mathcal{G} = (\boldsymbol{V}, \boldsymbol{E})$ be a DAG with the vertex set $\boldsymbol{V}$ and edge set $\boldsymbol{E}$. Consider $d$ observable variables denoted by $\boldsymbol{V} = (V_1, V_2, ..., V_d)$, and denote $\mathbb{P}$ as its probability distribution. From a statistical view, $X \perp\!\!\!\perp Y | Z$ denotes that $X$ and $Y$ are conditionally independent given $Z$. Moreover, from a graph view, $X \perp\!\!\!\perp_d Y | Z$ denotes that $X$ and $Y$ are d-separated by $Z$. Given $n$ data samples, the task of causal discovery aims at recovering the causal graph $\mathcal{G}$ from the data matrix $\boldsymbol{V} \in \mathbb{R}^{n \times d}$. Usually, each variable $V_i \in \boldsymbol{V}$ with random noises can be represented by the following structural causal model (SCM): $V_i = f_i(\mathrm{PA}_i, \epsilon_i)$, where $\mathrm{PA}_i$ is the set of all direct causes of $V_i$, and $\epsilon_i$ is the random noise with non-zero variance related to $V_i$, and we assume that $\epsilon_i$'s are mutually independent. For variables with deterministic relations, the SCM becomes: $V_i = f_i(\mathrm{PA}_i)$, where there is no extra noise. The relation can also be denoted as $\mathrm{PA}_i \mapsto V_i$, where $\mapsto$ is the deterministic function mapping, showing $\mathrm{PA}_i$ determines $V_i$. Throughout this paper, we assume causal sufficiency, i.e., no latent confounder.

**Terminologies.** Consider Figure 1(a) as an example, where $V_3$ has deterministic relation with $V_1$ and $V_2$, i.e., $V_3 = V_1 + V_2$, and $V_4$ is a non-deterministic variable. Here we call the set of deterministic variables as a *deterministic cluster (DC)*, e.g., $\{V_1, V_2, V_3\}$. Accordingly, all the non-deterministic variables make up a *non-deterministic cluster (NDC)*, e.g., $\{V_4\}$. Meanwhile, the edges connecting between DC and NDC compose a *bridge set (BS)*, e.g., $\{V_2 \rightarrow V_4, V_3 \rightarrow V_4\}$.

**Assumption 1 (Markov)** *Given a DAG $\mathcal{G}$ and the distribution $\mathbb{P}$ over the variable set $\boldsymbol{V}$, each variable is probabilistically independent of its non-descendants given its parents in $\mathcal{G}$.*

There are many DAGs that induce the same conditional independence relations with the distribution $\mathbb{P}$, and it is said to be Markov equivalent. The Markov equivalent class (MEC) contains all the DAGs which entail the same conditional independence relations as $\mathcal{G}$ does.

Another widely used assumption is faithfulness [7]. It states that any conditional independence that holds in the probability distribution must correspond to a d-separation in the causal graph. When the Markov and faithfulness assumptions hold true, constraint-based methods, such as PC, have been proven to output the correct MEC asymptotically. However, in the finite sample regime, the faithfulness assumption is sensitive to statistical testing errors when inferring the CI relations, and the violations might occur often. When there are deterministic relations, faithfulness also fails. Glymour [22] proposes the non-deterministic faithfulness regarding only non-deterministic variables. Moreover, relaxations of faithfulness have been proposed, such as adjacency-faithfulness [8] and triangle-faithfulness [9]. Another strictly weaker assumption is called Sparsest Markov Representation (SMR) [21], which is also known as the unique-frugality assumption [30, 31].

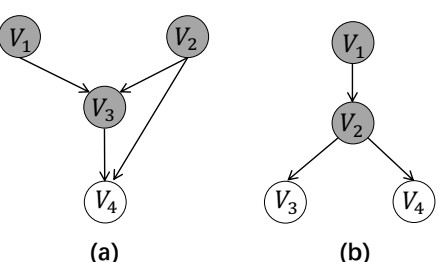

Figure 1: Two examples of causal graphs where faithfulness is violated. The gray nodes are deterministic variables. (a) $\{V_1, V_2\} \mapsto V_3$. Violation reason is $V_4 \perp\!\!\!\perp V_3 | \{V_1, V_2\}$ but $V_4 \not\perp\!\!\!\perp_d V_3 | \{V_1, V_2\}$. (b) $V_1 \mapsto V_2$. Violation reason is $V_3 \perp\!\!\!\perp V_4 | V_1$ but $V_3 \not\perp\!\!\!\perp_d V_4 | V_1$.

**Assumption 2 (Sparsest Markov Representation (SMR) [21])** *Given a DAG $\mathcal{G}$ and the distribution $\mathbb{P}$ over the variable set $\boldsymbol{V}$, the MEC of $\mathcal{G}$ is the unique sparsest MEC which satisfies the Markov assumption.*

The idea behind SMR is to find the sparsest graphical representation that captures the essential conditional independence relationships in the data. The term "sparsest" refers to the minimal number

of edges in the graphical model. Under the SMR assumption, the exact score-based methods, such as A* [15] and DP [13], can produce asymptotically correct results for learning the true MEC.

## 2.2 Faithfulness Violation by Deterministic Relations

Glymour [22] pointed out two and only two scenarios in the presence of deterministic relations where faithfulness can be violated. We summarize the two conditions and present the following assumption.

**Assumption 3 (Non-deterministic Faithfulness [22])** *Define a DAG $\mathcal{G}$ and the distribution $\mathbb{P}$ over the variable set $\boldsymbol{V}$. $\forall X, Y$ and $S$ in $\boldsymbol{V}$, if $X \perp\!\!\!\perp Y | S$ in $\mathbb{P}$ and none of the following conditions holds:*

    *i. $S \mapsto X$ or $S \mapsto Y$,*

    *ii. $\exists S'$ s.t. $X \perp\!\!\!\perp_d Y | S'$ and $S \mapsto S'$,*

*then $X \perp\!\!\!\perp_d Y | S$ in $\mathcal{G}$.*

**Remarks:** It assumes there is no other coincidental independence besides the two conditions. In other words, the two conditions are the only two cases leading to faithfulness violation due to deterministic relations. In fact, this assumption is equivalent to the completeness of D-separation criteria in Spirtes et al. [7]. We will use two graph examples, as shown in Figure 1, to explain the above two conditions.

Firstly, given condition (*i*) and Figure 1(a), we can assign $S = \{V_1, V_2\}$ and $X = V_3$, where $S \mapsto X$. Given $\{V_1, V_2\}$, $V_3$ will always be conditionally independent from $V_4$, because $V_3$ can be determined by $\{V_1, V_2\}$ with no extra noise term, the estimated residue for regressing $V_3$ on $\{V_1, V_2\}$ will be close to 0. Therefore, $V_4 \perp\!\!\!\perp V_3 | \{V_1, V_2\}$ holds true from a statistical view. However, $V_4 \not\perp\!\!\!\perp_d V_3 | \{V_1, V_2\}$ from a graph view. Therefore, in this case, faithfulness is violated.

The key rule of constraint-based method (e.g., PC algorithm) is that if we find at least one conditional set or an empty set so that two variables are conditionally independent, then the edge between these two variables in the graph will be removed. Therefore, we can conclude that using constraint-based methods which rely on faithfulness to deal with deterministic relations can be problematic.

Secondly, given condition (*ii*) and Figure 1(b), we can assign $S = V_1$, $S' = V_2$, $X = V_3$ and $Y = V_4$, where $S \mapsto S'$. From the graph, we can see that $V_3 \perp\!\!\!\perp_d V_4 | V_2$ and $V_3 \not\perp\!\!\!\perp_d V_4 | V_1$. However, from a statistical view $V_3 \perp\!\!\!\perp V_4 | V_2$, since $V_2 = V_1$, we also have $V_3 \perp\!\!\!\perp V_4 | V_1$. Here, conditional independence does not imply d-separation. Therefore, faithfulness is also violated.

## 2.3 Intuitive Solution: Exact Search

Benefiting from the recent theoretical progress on exact score-based methods, which do not explicitly rely on faithfulness assumption, it enables us to deal with deterministic relations from an intuitive view. Here, we are inspired by the lemma as follows.

**Lemma 1 (Linear Identifiability of Exact Search [20])** *Exact score-based search with BIC score asymptotically outputs a DAG that belongs to the MEC of the true DAG $\mathcal{G}$ if and only if the DAG $\mathcal{G}$ and distribution $\mathbb{P}$ satisfy the SMR assumption.*

According to Lemma 1, in the linear case, as long as the SMR assumption is satisfied, the exact score-based method with BIC score [11] can asymptotically obtain the true MEC. Then, we can extend the theoretical result from linear to nonlinear scenarios. The exact score-based method with generalized score [32] can also asymptotically output the true MEC.

**Theorem 2 (General Identifiability of Exact Search)** *Exact score-based search with generalized score asymptotically outputs a DAG that belongs to the MEC of the true DAG $\mathcal{G}$ if and only if the DAG $\mathcal{G}$ and distribution $\mathbb{P}$ satisfy the SMR assumption and some mild conditions are satisfied.*

**Remarks:** The complete proof is given in Appendix A4.1. Based on the theoretical findings in Lemma 1 and Theorem 2, the exact score-based methods, which do not specifically require faithfulness but SMR, pave a promising way to deal with the deterministic relations for causal discovery. However, one critical disadvantage of the exact methods is their low computational efficiency and poor scalability. To that end, we propose a novel framework, called DGES, which is demonstrated in section 3. The identifiability conditions for DGES are provided in section 4.

---

**Algorithm 1** DGES: **D**eterminism-aware **G**reedy **E**quivalent **S**earch

---

**Input:** data matrix $\mathcal{D} \in \mathbb{R}^{n \times d}$
**Output:** a causal graph $\mathcal{G}$
 1: *(Phase 1: Detect Minimal Deterministic Clusters)* Detect the minimal deterministic clusters, by checking whether one variable can be minimally determined by some other variables.
 2: *(Phase 2: Run Modified Greedy Search Globally)* Run modified greedy equivalent search on the whole set of variables to obtain an initial graph.
 3: *(Phase 3: Run Exact Search Partially)* Perform the exact search exclusively on the deterministic clusters and their neighboring variables, as post-processing.

---

## 3 Determinism-aware Greedy Equivalent Search (DGES)

In this section, we will introduce our proposed DGES in detail. Throughout this paper, we consider the general case without assuming any functional causal models. In general, DGES contains three phases: Firstly, we need to detect all the minimal deterministic clusters. If one variable can be deterministically represented by some other variables, we may conclude that it is a deterministic variable. Secondly, based on the DC information, we run modified GES to get the initial causal graph. Thirdly we perform the exact search exclusively on the DC and their neighbors, as post-processing. The general framework is given in Algorithm 1. The contents are organized as follows. The details of deterministic cluster detection in Phase 1 are discussed in section 3.1. More information about our modified GES in Phase 2 is introduced in section 3.2. Finally, we discuss exact search in section 3.3.

### 3.1 Minimal Deterministic Clusters Detection

A *minimal deterministic cluster (MinDC)* refers to a minimal set of variables involved in a deterministic relation. A DC can be seen as a union of all MinDCs in the graph. For example, $V_1 \mapsto V_2$ and $V_1 \mapsto V_3$, then $\{V_1, V_2\}$ and $\{V_1, V_3\}$ are two MinDCs, while $\{V_1, V_2, V_3\}$ composes a DC.

First of all, we need to obtain the DC, which contains all the deterministic variables. For each variable $V_i, i \in \{1, ..., d\}$, if this variable can be deterministically represented by all the other variables, i.e., $\{\boldsymbol{V} \backslash V_i\} \mapsto V_i$, then this variable must be in DC. After traversing all $d$ variables, we obtain the DC.

However, within the DC, there may be multiple deterministic relations, even some overlapping deterministic variables. Therefore, out of the DC, we need to get a set of MinDCs. For each variable $V_i$, we try to detect whether there exists a minimal set $S$ such that $S \mapsto V_i$, where $V_i \in \mathrm{DC}, S \subset \mathrm{DC}$ and $V_i \notin S$. Here, we need to traverse all the possible combination sets of DC, and see whether one deterministic variable can be minimally represented by some other variables. If so, then those variables compose a MinDC. In the end, we can obtain a list of MinDCs. More details about DC detection, MinDC detection, and how to check $S \mapsto V_i$, are given in Appendix A3.1.

### 3.2 Modified Greedy Equivalent Search

The modified GES is based on the standard GES [10]. We add some extra constraints during the forward and backward steps and adjust the score functions due to the deterministic relations. When using score functions for causal discovery, we aim for the underlying causal graph or its equivalent class to give the optimal score. Specifically, we desire that the score of a DAG model (1) increases as the result of adding any edge that eliminates an independence constraint that does not hold in the generative distribution, and (2) decreases as a result of adding any edge that does not eliminate such a constraint. More formally, we have the following definition of score local consistency.

**Definition 1 (Score Local Consistency [10])** *Let $\mathcal{G}$ be any DAG, and let $\mathcal{G}'$ be the DAG that results from adding the edge $V_i \to V_j$ on $\mathcal{G}$. Let $D$ be the dataset from the distribution $\mathbb{P}$. A score function $\mathbb{S}(\mathcal{G}; D)$ is locally consistent if the following two properties hold as the sample size $n \to \infty$:*

    *1. If $V_i \not\perp\!\!\!\perp V_j | PA_j^{\mathcal{G}}$, then $\mathbb{S}(\mathcal{G}'; D) > \mathbb{S}(\mathcal{G}; D)$.*

    *2. If $V_i \perp\!\!\!\perp V_j | PA_j^{\mathcal{G}}$, then $\mathbb{S}(\mathcal{G}'; D) < \mathbb{S}(\mathcal{G}; D)$.*

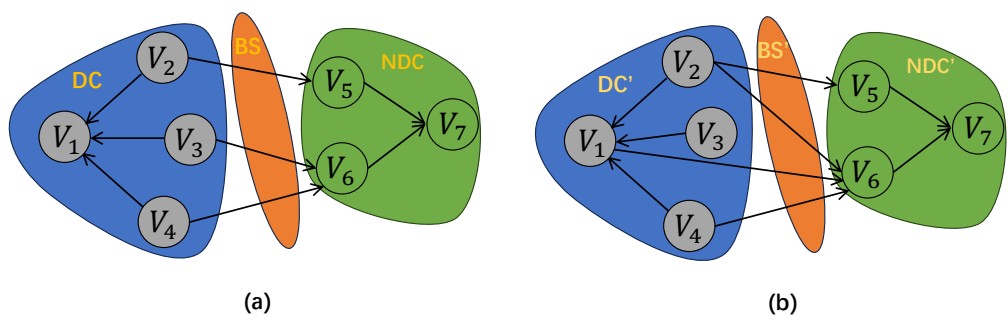

Figure 2: An example graph where $V_1 = f(V_2, V_3, V_4)$. (a) the true graph where DC = $\{V_1, V_2, V_3, V_4\}$, NDC = $\{V_5, V_6, V_7\}$, and BS = $\{V_2{\rightarrow}V_5, V_3{\rightarrow}V_6, V_4{\rightarrow}V_6\}$. (b) one possible DAG from the estimated CPDAG by GES, where BS' = $\{V_2{\rightarrow}V_5, V_1{\rightarrow}V_6, V_2{\rightarrow}V_6, V_4{\rightarrow}V_6\}$

**Modification 1: Edge Adding and Deleting.** During the forward phase, at each step with a DAG $\mathcal{G}$ in the equivalence class, an edge $V_i \rightarrow V_j$ is added when 1) $V_i \not\perp\!\!\!\perp V_j | \mathrm{PA}_j^{\mathcal{G}}$, and 2) $\mathrm{PA}_j^{\mathcal{G}}$ does not determine any of $V_i, V_j$, until no edge can be added. However, when $\mathrm{PA}_j^{\mathcal{G}}$ determines $V_i$ or $V_j$, we always have $V_i \perp\!\!\!\perp V_j | \mathrm{PA}_j^{\mathcal{G}}$. In this case, we always ignore such independence, directly regard it as dependent, and add such an edge to the graph. The motivation behind the modification is to ensure that no false independence due to deterministic relations is introduced, and in the end, the output graph is guaranteed to be Markovian.

During the backward phase, at each step with a DAG $\mathcal{G}$ in the equivalence class, an edge $V_i \rightarrow V_j$ is removed when both 1) $V_i \perp\!\!\!\perp V_j | \mathrm{PA}_j^{\mathcal{G}}$, and 2) $\mathrm{PA}_j^{\mathcal{G}}$ does not determine any of $V_i, V_j$, until no edge can be removed. Similar to the modification in the forward phase, when $\mathrm{PA}_j^{\mathcal{G}}$ determines $V_i$ or $V_j$, we still trust the dependency and keep the edge $V_i \rightarrow V_j$. Although the resulting equivalence class will be Markovian to the ground truth, redundant edges will exist.

Fortunately, we have Phase 3 exact search as post-processing, which will be introduced next in Section 3.3. Under the SMR assumption, we can obtain a more sparse graph. In the end, the exact search will remove all those redundant edges. A motivating example showing the advantages of our modified forward and backward phases is provided in Appendix A3.2 and Figure A2.

**Modification 2: Score Function.** During the phase 1 with greedy search and phase 3 with exact search, a proper score function is inevitably needed. For any scoring criterion $\mathcal{S}(\mathcal{G}, \mathcal{D})$, we say that a score is *decomposable* if it can be written as a sum of local scores, where each local score is a function of only one variable and its parents. Following the property, the score of a DAG $\mathcal{G}$ can be represented as

$$\mathbb{S}(\mathcal{G}; \mathcal{D}) = \sum_{i=1}^{d} \mathcal{S}(V_i, \mathrm{PA}_i^{\mathcal{G}}). \tag{1}$$

Under the linear Gaussian model, the BIC score [11] is preferred, which is given as

$$\mathcal{S}_{BIC}(V_i, \mathrm{PA}_i^{\mathcal{G}}) = -\log L + \lambda' k \log n,$$
$$and \ \ \log L \propto -\frac{n}{2}(1 + \log|\Sigma|), \tag{2}$$

where $L$ is the maximized value of the likelihood function of the model based on the observed data $\mathcal{D}$ related to $V_i$ and $\mathrm{PA}_i$, $k$ denotes the number of edges between $V_i$ and $\mathrm{PA}_i$ in $\mathcal{G}$, $n$ is the number of data samples in $\mathcal{D}$, $\lambda'$ is the penalty parameter, $\Sigma$ is the variance of the noise term.

However, in the deterministic scenarios, the estimated noise variance $\hat{\Sigma}$ will asymptotically get closer to 0, which leads to numerical error because of the term $\log|\hat{\Sigma}|$. To deal with such an issue, we provide the adjusted BIC score, formulated as

$$\mathcal{S}'_{BIC}(V_i, \mathrm{PA}_i^{\mathcal{G}}) = -\log L' + \lambda' k \log n,$$
$$and \ \ \log L' \propto -\frac{n}{2}(1 + \log|\Sigma + \xi|), \tag{3}$$

where $\xi$ is a small constant, and $\xi > 0$.

Under the general nonlinear model, the generalized score (GS) [32] which is in a non-parametric form is favored. There are two types of likelihoods as introduced in the paper, for computational efficiency, we choose the generalized score with cross-validated (CV) likelihood.

$$\mathcal{S}_{GS}(V_i, \mathrm{PA}_i^{\mathcal{G}}) = \frac{1}{Q} \sum_{q=1}^{Q} \ell(F_i^{(q)} | D_{0,i}^{(q)}), \quad and$$

$$
\begin{aligned}
\ell(\hat{F}_i^{(q)} | D_{0,i}^{(q)}) = & -\frac{n_0^2}{2} \log(2\pi) - \frac{n_0}{2} \log \left| n_1 \lambda^2 \tilde{K}_{V_i}^{1(q)} (\tilde{K}_{PA_i^{\mathcal{G}}}^{1(q)} + n_1 \lambda I)^{-2} \tilde{K}_{V_i}^{0(q)} \right| \\
& - \frac{1}{2} \mathrm{trace} \Big\{ \frac{1}{\lambda} \tilde{K}_{V_i}^{0(q)} \tilde{K}_{V_i}^{0(q)} + \frac{1}{\lambda} \tilde{K}_{PA_i^{\mathcal{G}}}^{0,1(q)} A_i^{\mathrm{T}} A_i \tilde{K}_{PA_i^{\mathcal{G}}}^{1,0(q)} - n_1 \tilde{K}_{PA_i^{\mathcal{G}}}^{0,1(q)} A_i^{\mathrm{T}} B_i A_i \tilde{K}_{PA_i^{\mathcal{G}}}^{1,0(q)} \\
& + 2n_1 \tilde{K}_{V_i}^{0(q)} B_i A_i \tilde{K}_{PA_i^{\mathcal{G}}}^{1,0(q)} - \frac{2}{\lambda} \tilde{K}_{V_i}^{0(q)} A_i \tilde{K}_{PA_i^{\mathcal{G}}}^{1,0(q)} - n_1 \tilde{K}_{V_i}^{0(q)} B_i \tilde{K}_{V_i}^{0(q)} \Big\},
\end{aligned}
\tag{4}
$$

where $A_i = \tilde{K}_{V_i}^{1(q)} (\tilde{K}_{PA_i^{\mathcal{G}}}^{1(q)} + n_1 \lambda I)^{-1}$, $B_i = A_i (I + n_1 \lambda A_i^{\mathrm{T}} A_i)^{-1} A_i^{\mathrm{T}}$, $\lambda$ is the regularization parameter, $n_1$ is the sample size of each training set, $n_0$ is the sample size of each test set, $n = n_1 + n_0$, $D_{1,i}^{(q)}$ and $D_{0,i}^{(q)}$ are the corresponding data of variable $V_i$ and its parents, $\tilde{K}_{V_i}^{1(q)}$ denotes the centralized kernel matrix of the $q$-th training set of $V_i$, $\tilde{K}_{V_i}^{0(q)}$ denotes that of the $q$-th test set of $V_i$, and similar notations are used for other kernel matrices.

### 3.3 Exact Search as Post-processing

As demonstrated by Lu et al. [19], GES may get sub-optimal results when the faithfulness assumption is violated, e.g., when there are deterministic relations. An example is given in Figure 2. In this example, the DC is $\{V_1, V_2, V_3, V_4\}$. The true incoming edges to $V_6$ should be $\{V_3, V_4\}$, however, the estimated graph by GES may have $\{V_1, V_2, V_4\}$ pointing to $V_6$. We need to partially conduct an exact search based on the GES result to identify BS, under the SMR assumption. Therefore, in Phase 3, we perform the exact search exclusively on the DC and their neighbors. Benefiting from the recent theoretical progress on exact score-based methods, which do not explicitly rely on faithfulness assumption, it enables us to deal with deterministic relations from an intuitive view.

## 4 Identifiability Conditions

In this section, we provide the identifiability conditions of DGES. The conditions are presented in a general form, applicable to both linear and nonlinear causal models. As mentioned above, in a general deterministic system, the whole causal graph mainly can be divided into three parts: DC, NDC, and BS. In this paper, we focus on the identifiability for the BS and NDC parts.

**Theorem 3 (Partial Identifiability)** *Denote a causal graph $\mathcal{G}$ with deterministic relations. Let $V_i$ be any non-deterministic variable in $\mathcal{G}$, and $\mathrm{PA}_i$ be the set of direct causes or undirected neighbors of $V_i$ in one MinDC. Suppose the following conditions hold*

    *i. Assumptions 1, 2, and 3 hold,*

    *ii. $|\mathrm{PA}_i| < |\mathrm{MinDC}| - 1$,*

*where $|\cdot|$ denotes the cardinality of a set. Then, when the sample size $n \to \infty$, we can identify the BS and NDC parts of the causal graph $\mathcal{G}$ to their true Markov equivalent class.*

## 5 Experiments

To validate our theoretical findings and show the efficacy of our method, we conducted extensive experiments on simulated and real-world datasets. Specifically, for simulated datasets, we evaluate both linear and general nonlinear functional models.

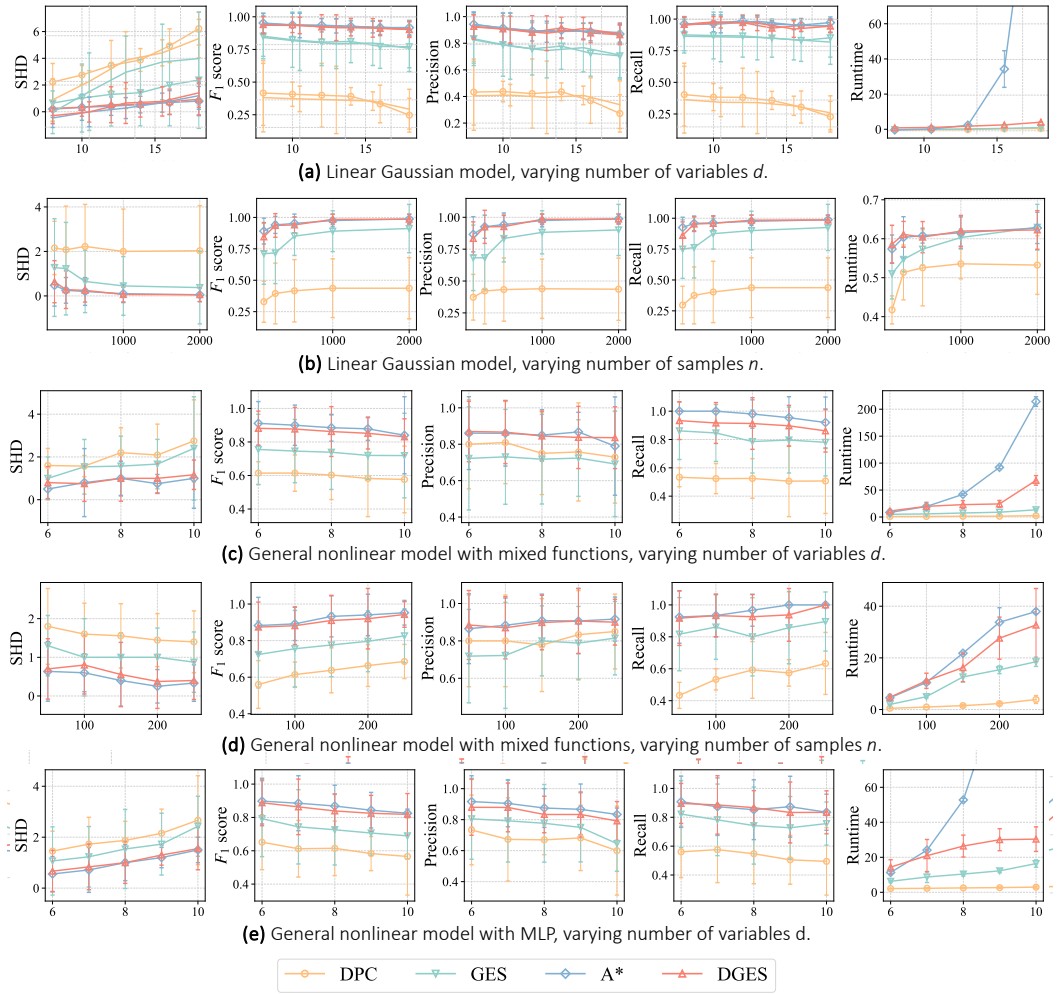

(a) Linear Gaussian model, varying number of variables $d$.

(b) Linear Gaussian model, varying number of samples $n$.

(c) General nonlinear model with mixed functions, varying number of variables $d$.

(d) General nonlinear model with mixed functions, varying number of samples $n$.

(e) General nonlinear model with MLP, varying number of variables d.

| DPC | GES | A* | DGES |

Figure 3: Results on the simulated datasets with one MinDC. We evaluate different functional causal models on varying number of variables and samples, respectively. For each setting, we consider SHD ($\downarrow$), $F_1$ score ($\uparrow$), precision ($\uparrow$), recall ($\uparrow$) and runtime ($\downarrow$) as evaluation criteria.

**Simulated Datasets.** The true DAGs are simulated using the Erdös–Rényi model [33] with the number of edges equal to the number of variables. We evaluate linear Gaussian model and general nonlinear model with mixed functions, each with varying number of variables and samples. Moreover, we also evaluate general nonlinear model generated by MLP on varying number of variables. For each setting, we randomly choose one MinDC or two MinDCs where each MinDC has at least three variables. For the exact method in Phase 3, we choose A* [15] without the heuristic tricks. We compare our DGES with other baselines, including DPC [22], GES [10], and A* [15]. We compare the MEC of the output by all methods. Note that we only evaluate the BS part which we aim to identify. We consider the structural Hamming distance (SHD), the $F_1$ score, the precision, the recall, and the computational time as evaluation criteria. For each setting, we run 10 different random seeds and report the mean and standard deviation. More implementation details are in Appendix A5.1.

The simulated results about graphs with only one DC has been shown in Figure 3, and the results with two DCs (which may have overlapping variables) are given in Figure A4 of Appendix. Clearly, when there are more deterministic variables in the system, the runtime of our DGES will obviously increase. The reason is because there are more deterministic variables to be detected and fed into Phase 3 for exact search. According to the results, the general performance of DGES is competitive compared to other baselines. We observe that the exact method A* and our proposed DGES generally outperform the other baselines such as GES and DPC across different criteria and settings. Meanwhile, score-based method GES presents better performance than constraint-based method DPC in a deterministic

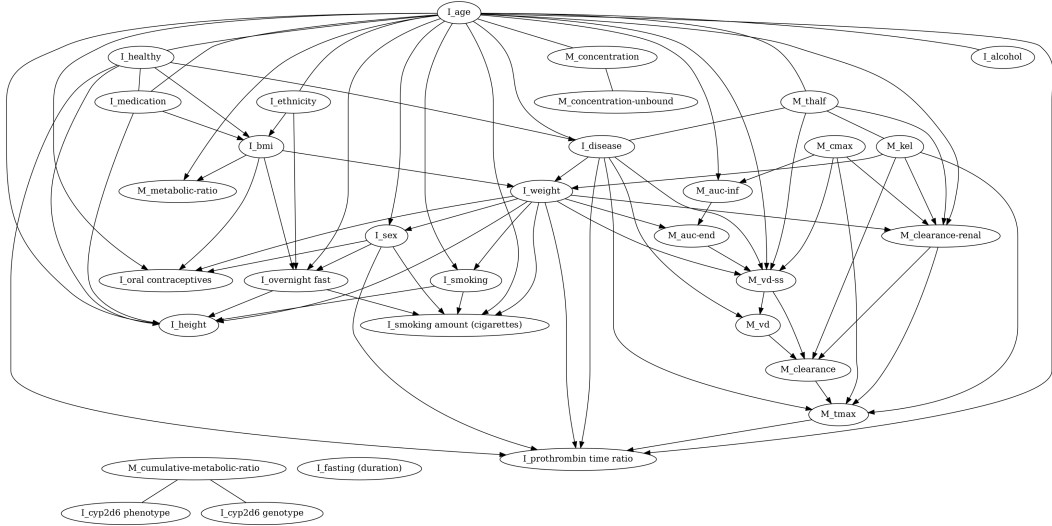

Figure 4: Results on the real-world dataset with deterministic relations by DGES with Generalized score.

system. As the number of variable increases, the runtime of A* will increase rapidly. Compared to A*, the increasing of runtime for DGES is much more steady, both in linear and nonlinear models. More results about two MinDCs, non-deterministic scenarios, and relaxed exact search such as GRaSP [31], are provided in Appendix A5.

**Real-world Datasets.** We also evaluate our method and the baselines on two real-world datasets. One is the pharmacokinetics dataset [34], which is an open database for pharmacokinetics information from clinical trials. It provides curated information mainly in two categories: the characteristics of the studied individuals (e.g., age, height) and the measurement records (e.g., the clearance, $T_{max}$, $C_{max}$ when one certain individual takes one certain drug), and we name the two categories of variables as class "I" (individual) and "M" (measurement), respectively. Out of more than 200 variables and more than 200000 data samples containing missing values, we cleaned the data and finally obtained 32 important variables with 4194 data samples which may contain deterministic relations. The 32 variables contains 18 and 14 variables from the class "I" and "M", respectively. We prepend the class label to each variable name as a prefix. We use linear BIC score and nonlinear generalized score to conduct the search. Figure 4 gives the DGES result with generalized score, where we can successfully detect at least three MinDCs: {height, weight, BMI}, {$k_{el}$, $V_d$, Clearance}, {$k_{el}$, $T_{half}$}. Compared with the linear DGES result with BIC score, we can see more reasonable edges existing in the nonlinear DGES result with the generalized score, for example, {age − medication, healthy → disease, healthy − BMI}. More results and analysis are provided in Appendix A6.

The other one is the US census Public Use Microdata Sample (PUMS). We follow the data preprocessing procedure outlined in [35], which is a modern version of the UCI Adult data set [36]. Datasets based on census data are widely considered in the algorithmic fairness literature [37–41]. Here we choose 5 important variables, i.e., Age, Occupation, Sex, Annual income (AI), and Adjusted annual income (AAI), in total there are 3000 samples. Because of the potentially different timeframe of the survey cycle, AAI (= AI * Adjusted factor) are the adjusted dollar amounts that they have earned entirely during the calendar year. Within one calendar year, this adjusted factor is a constant. Here, we choose the data in 2021. Therefore, AAI and AI have a deterministic relation. The result of DGES is: {Sex → Occupation ← AI, AI ← AAI, Sex → AAI ← Age}, 5 edges. The result of GES is: {Sex ← Occupation – AAI, AI – AAI – Age, AI → Sex ← AAI}, 6 edges. The result of PC is: {Sex → Occupation ← Age, AI – AAI}, 3 edges. Compared with GES, the result of DGES is more sparse. Particularly, we can detect that AI and AAI have a deterministic relation, and GES gives redundant edges by {AI → Sex ← AAI} while our DGES only keeps one edge {Sex → AAI}. Moreover, the result of PC is totally different from the other two. Clearly, in our DGES result, AI and AAI are still connected, and we can still see the BS, i.e., {AI → Occupation, AAI – Sex, AAI–Age}. However, as a result of PC with FisherZ test, the BS becomes empty, which is exactly due to the violation of faithfulness.

# 6 Discussions and Conclusion

**Limitations.** While presenting a versatile framework, our paper does have certain limitations. Firstly, in some cases, e.g., with overlapping deterministic variables, our method cannot identify the skeleton and directions in the DC part so far. We display two graphs that we can identify up to MEC and the other two that we cannot identify in Figure A1. More discussion is in Appendix A1. Secondly, inherited from the disadvantages of exact methods, our method can be somewhat computationally expensive in Phase 3 when there are a large number of MinDCs. Fortunately, each MinDC is usually not too large, and we may execute the exact search for different MinDCs simultaneously.

**Broader Impacts.** The overarching aim of our proposed method is to learn the causal structures from any general functional causal models in the presence of deterministic relations. This is a fundamental and critical task with wide-ranging applications in practical life, and we firmly believe that our method will serve beneficial purposes without engendering negative societal impacts.

**Conclusion.** This paper dives into the challenges of causal discovery in the presence of deterministic relations. Notably, we make a compelling discovery that exact score-based methods can elegantly address the deterministic issues, provided the SMR assumption is met. In an effort to bolster efficiency and scalability in a deterministic system, we propose the novel and versatile framework called DGES, encompassing both linear and nonlinear models, as well as both continuous and discrete data types. Furthermore, we establish the partial identifiability conditions for DGES. Hopefully, our method can help to construct a holistic view to see the deterministic relations. The extensive experiments on simulated and real-world datasets, validate our theoretical findings and the efficacy of our method.

## Acknowledgement

This material is based upon work supported by NSF Award No. 2229881, AI Institute for Societal Decision Making (AI-SDM), the National Institutes of Health (NIH) under Contract R01HL159805, and grants from Salesforce, Apple Inc., Quris AI, and Florin Court Capital.

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

*Appendix for*

## "On Causal Discovery in the Presence of Deterministic Relations"

Table of Contents:

## A1   More Discussions

**Q1: Why current method cannot identify the skeleton and directions in the DC part?**

**A1:** To achieve that goal, we usually need strong assumptions on the underlying functional causal model, i.e., Yang et al. [29] assumed linear non-Gaussian model. However, those assumptions are not in alignment with our goal of a general method, i.e., with no restricted assumption on the underlying functional causal model. That is why currently our method cannot identify the skeleton and directions in the DC part. However, fortunately, we can exactly find out which set of variables are in the DC/MinDCs using some DC detection strategies, as shown in Section 3.1.

Figure A1 gives three example graphs where two of them can be identified up to the true MEC while the other one cannot. In graph (a), $V_1 \mapsto V_2$, after DGES, we can capture the dependence between $V_1$ and $V_2$, therefore, we can identify in this case. Similarly, in the graph (b), $\{V_1, V_3\} \mapsto V_2$, since $V_1$ and $V_3$ are independent, GES can still capture this v-structure. Therefore, we can identify in this case. However, things are different in the two examples at right. In graph (c), $V_1 \mapsto V_2$, $V_2 \mapsto V_3$, after running GES, we may get a fully-connected graph. Obviously, this fully-connected graph has different skeleton and directions than the true one.

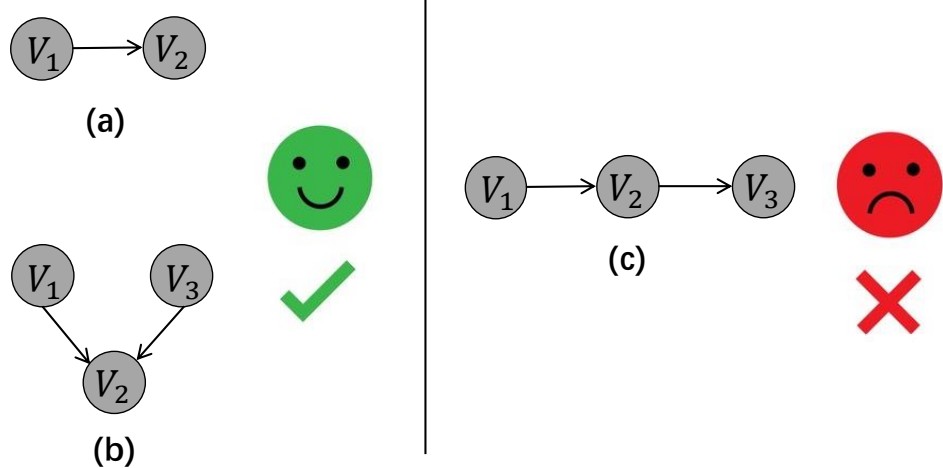

Figure A1: Some graphs where DGES can (**Left**) or cannot (**Right**) identify the MEC: (a) $V_1 \mapsto V_2$, (b) $\{V_1, V_3\} \mapsto V_2$, (c) $V_1 \mapsto V_2, V_2 \mapsto V_3$.

**Q2: Why GES can be problematic in the cases where DC variables cause NDC variables?**

**A2:** Take Figure 2 as an example, where the true edges related to $V_6$ include $V_3 \to V_6$ and $V_4 \to V_6$. However, during the forward phase of GES, it is very likely that the edge $V_1 \to V_6$ can be added in the beginning. Then the edges $V_2 \to V_6$ and $V_4 \to V_6$ are added subsequently. During the backward phase of GES, the edge $V_1 \to V_6$ will not be deleted, because $V_1$ also contains information from $V_3$, in other words, $V_6$ is represented by $V_1$, $V_2$ and $V_4$ by GES, which contains more edges than the ground-true. Therefore, in this case GES can be problematic, and we need exact search under the SMR assumption for post-processing, in order to correctly identify the BS part.

**Q3: How GES performs in the cases where NDC variables cause DC variables?**

**A3:** Let's consider this example. Three variables compose a DC ($V_1, V_2, V_3$, and $V_1 = V_2 + V_3$), and denote another variable from NDC as $V_4$. In this case, there must not be an edge from $V_4$ to $V_1$, because $V_4$ will be in DC rather than NDC, if so. Then, if $V_4$ causes $V_2$, there will definitely be an edge between them by GES because $V_4$ is clearly dependent on $V_2$, and theoretically, GES can capture this dependence based on the local consistency.

**Q4: What the characterization of Markov equivalence class is in the context with deterministic relations?**

**A4:** Regarding only the variables involved in BS and NDC (that is how Theorem 4 claimed), the characterization of Markov equivalence class (MEC) with deterministic relations is still the same as the context of not having deterministic relations.

However, if we consider the whole graph, i.e., all of the variables in DCs are also involved, the characterization of the Markov equivalence class should be different. As shown in the condition (*i*) of Assumption 3 and Figure 1, there will be "constant independence" caused by the deterministic relations. Therefore, we need to remove those "constant independence" for the new characterization of MEC.

## A2 Related Works

In this part, we will introduce more related works in causal discovery [42]. As we mentioned in the main paper, constraint-based and score-based methods are two primary categories in causal discovery. Constraint-based methods utilize the conditional independence test (CIT) to learn a skeleton of the directed acyclic graph (DAG), and then orient the edges upon the skeleton. Such methods contain Peter-Clark (PC) algorithm [42] and Fast Causal Inference (FCI) algorithm [43]. Some typical CIT methods include kernel-based independent conditional test [44] and approximate kernel-based conditional independent test [45, 46].

---
**Algorithm A1** Detecting Deterministic Cluster (DC)
---
**Input:** variable set $\boldsymbol{V} \in \mathbb{R}^d$
**Output:** DC
 1: DC $\leftarrow \emptyset$
 2: **for** $i = 1$ to $d$ **do**
 3:     **if** $\{\boldsymbol{V} \backslash V_i\} \mapsto V_i$ **then**
 4:         DC $\leftarrow$ DC $\cup \{V_i\}$
 5:     **end if**
 6: **end for**
---

Score-based methods normally use a score function and rely on a particular search strategy to look for the intended graph. The search strategy usually involve greedy search, exact search, or continuous optimization. The first continuous-optimization based method is NOTEARS [47], which casts the Bayesian network structure learning task into a continuous constrained optimization problem with the least squares objective, using an algebraic characterization of directed acyclic graph (DAG) [48]. Subsequent work GOLEM [49] adopts a continuous unconstrained optimization formulation with a likelihood-based objective. NOTEARS is designed under the assumption of the linear relations between variables, therefore, another subsequent works have extended NOTEARS to handle nonlinear cases via deep neural networks, such as DAG-GNN [50] and DAG-NoCurl [51]. Moreover, ENCO [52] presents an efficient DAG discovery method for directed acyclic causal graphs utilizing both observational and interventional data. AVCI [53] infers causal structure by performing amortized variational inference over an arbitrary data-generating distribution. These methods might suffer from various optimization issues, including convergence [54], sensitivity to data standardization [55], and nonconvexity [56]. Since they are only guaranteed to find a local optimum, therefore the quality of the solution can not be guaranteed, even in the asymptotic cases.

Besides the constrain-based and score-based methods, another major category of causal discovery methods is function causal model based methods. Those methods rely on the causal asymmetry property, such as the linear non-Gaussian model (LiNGAM) [57], the additive noise model [58], and the post-nonlinear causal model [59]. Apart from those methods, there are also some hybrid methods, such as neural conditional dependence (NCD) method, which reframes the GES algorithm to be more flexible than the standard score-based version and readily lends itself to the nonparametric setting with a general measure of conditional dependence.

**deterministic relations and faithfulness violation.** It is interesting to discuss the relationships between deterministic relations and faithfulness violation. These faithfulness relaxation methods such as [60] work on general faithfulness violation and propose some weaker faithfulness assumptions. They usually focus on certain types of structure, such as canceling path, XOR-type, triangle faithfulness, etc. However, to the best of our knowledge, deterministic relations will break all those relaxed faithfulness assumptions, as the distribution is even not a graphoid. Therefore, we need to develop specific algorithms to handle determinism.

## A3 Method Details

### A3.1 Phase 1: Minimal Deterministic Clusters Detection

**DC Detection.** In order to detect the DC, which contains all the deterministic variables, we need to traverse all $d$ variables. If $\{\boldsymbol{V} \backslash V_i\} \mapsto V_i$ holds true, then this variable $V_i$ must be in DC. The general pseudocode is stated in Algorithm A1.

**MinDCs Detection.** We aim to get a set of MinDCs from the DC obtained above. Given the DC, for each variable $V_i$, we try to detect whether there exists a minimal set $S$ such that $S \mapsto V_i$, $S \subset$ DC. As shown in the Algorithm A2, we need to traverse all the possible sets for $S$ with increasing cardinality $k$, $|S| = k$ ($|\cdot|$ means the cardinality of a set). If we find that $S \mapsto V_i$ and meanwhile $\{S \cup V_i\}$ is not a superset of any MinDC in current MinDCs, then we can conclude that $S \cup V_i$ composes one MinDC. Otherwise, if we find that $\{S \cup V_i\}$ is a superset of one MinDC $M$ in current MinDCs, we may conclude that $\{S \cup V_i\}$ is not a minimal DC because $|S \cup V_i| > |M|$.

---
**Algorithm A2** Detecting Minimal Deterministic Clusters (MinDCs)
---
**Input:** DC
**Output:** MinDCs
 1: MinDCs $\leftarrow \emptyset$
 2: **for** $k = 1$ to $|\,\mathrm{DC}\,| - 1$ **do**
 3:   **for** $i = 1$ to $|\,\mathrm{DC}\,|$ **do**
 4:     **for** each $S$ in $Combination(\mathrm{DC} \setminus V_i,\ k)$ **do**
 5:       **if** $S \mapsto V_i$ **then**
 6:         MinDC $\leftarrow S \cup \{V_i\}$
 7:         **if** $\forall M \in$ MinDCs s.t. $M \not\subset$ MinDC **then**
 8:           MinDCs $\leftarrow$ MinDCs $\cup$ {MinDC}
 9:         **end if**
10:       **end if**
11:     **end for**
12:   **end for**
13: **end for**
---

**How to Evaluate $S \mapsto X$?**  Here we use regression and evaluate the variance term of residue to decide whether there is a deterministic relation or not. Please note that DGES does not assume any functional causal model. Therefore, we also evaluate it in a general form. Specifically, we provide two versions: one assumes a linear model and is based on linear regression as shown in Lemma 4, and another is based on a general non-linear model as exhibited in Lemma 5.

**Lemma 4 (Representation in Linear Model)**  *Let $X$ be a random variable and $S$ be a set of random variables, where $X \notin S$. Define $X$ and $S$ are with domain $\mathcal{X}$ and $\mathbb{S}$, respectively. Consider a linear regression framework: $\mathcal{X}(X) = a * S + u$, where $a$ and $u$ represent the regression coefficient and residue, respectively.*

*$X$ can be represented by $S$ if and only if*

$$\mathrm{Var}(u) = 0, \tag{5}$$

*where $\mathrm{Var}(u)$ is the variance of the residue $u$.*

**Lemma 5 (Representation in General Nonlinear Model)**  *Let $X$ be a random variable and $S$ be a set of random variables, where $X \notin S$. Define $X$ and $S$ are with domain $\mathcal{X}$ and $\mathbb{S}$, respectively. Define a RKHS $\mathcal{H}_{\mathcal{X}}$ on $\mathcal{X}$ with continuous feature mapping $\phi_{\mathcal{X}} : \mathcal{X} \to \mathcal{H}_{\mathcal{X}}$. Consider a regression framework in the RKHS: $\phi_{\mathcal{X}}(X) = F_{\mathbb{S}}(S) + u$, where $F_{\mathbb{S}} : \mathcal{S} \to \mathcal{H}_{\mathcal{X}}$ and $u$ represents the regression residue.*

*$X$ can be represented by $S$ if and only if*

$$\|\Sigma_u\|_{HS}^2 = 0, \tag{6}$$

*where $\Sigma_u$ is the variance matrix of the residue, $\Sigma_u = R_u^T R_u$, $R_u = \varepsilon(\boldsymbol{K}_S + \varepsilon I)^{-1}\phi(X)$, $\varepsilon$ is a small positive regularization parameter for kernel ridge regression, and $\boldsymbol{K}_S$ is the centralized kernel matrix of $S$.*

### Discussion: Why consider kernel regression in Lemma 5?

Because we are considering the general functional causal form. Particularly, this Lemma can be used for both linear and nonlinear functional relationships, Gaussian and non-Gaussian data distributions, which is in alignment with the general goal of our proposed method. For more details, inspired by [44], the functions $\phi_{\mathcal{X}}$ and $F(\cdot)$ that we use are all in the infinite Hilbert spaces, and we evaluate the representation with the Hilbert-Schmidt norm of the variance operator $\Sigma_u$ in infinite dimension. In this case, we can exhibit a general functional causal form.

### Proof of Lemma 5:

Assume there is a MEC $\mathcal{M}$, which contains both directed edges and undirected edges. Let $X$ be a random variable in $\mathcal{M}$ and $S$ be the set of all non-descendant neighbors, including direct causes and undirected neighbors of $X$. Suppose the random variables $X$ and $S$ are over measurable spaces $\mathcal{X}$ and $\mathcal{S}$, respectively.

Without assuming a particular functional causal form, we usually exploit a regression framework in the RKHS, to encode general dependence relations between two random variables. Define a RKHS $\mathcal{H}_{\mathcal{X}}$ on $\mathcal{X}$ with continuous feature mapping $\phi_{\mathcal{X}} : \mathcal{X} \to \mathcal{H}_{\mathcal{X}}$. Here, we consider

$$\phi_{\mathcal{X}}(X) = F(S) + u, \tag{7}$$

where $F : \mathcal{S} \to \mathcal{H}_{\mathcal{X}}$ and $u$ represents the regression residue or noise. When applying the kernel ridge regression, we can obtain the estimated residue

$$\hat{u} = \varepsilon(\boldsymbol{K}_Z + \varepsilon I)^{-1}\phi(X), \tag{8}$$

where $\varepsilon$ is a small positive regularization parameter for kernel ridge regression, and $\boldsymbol{K}_Z$ is the centralized kernel matrix of $Z$. To evaluate whether such a residue exists, one may consider the Hilbert-Schmidt norm of the variance matrix

$$\|\Sigma_u\|_{HS}^2 = \|\hat{u}^T\hat{u}\|_{HS}^2 = 0, \tag{9}$$

If the above equation holds true, then we may conclude that there is no noise term in the relationship between $X$ and $S$, in other words, $X$ can be represented by $S$ (without extra noise term).

Vice versa.

### A3.2 Phase 2: Modified Greedy Equivalent Search

Figure A2 presents an example comparing our modified GES with traditional GES. From this example, we can see that: in the backward phase, if we use the "constant independence" information $C \perp\!\!\!\perp A|D$, then the result graph will become totally wrong where A and B will be connected. In our modifies GES, we indifferently ignore such "constant independence" information. In the end, some other information will be considered as a priority. For example, as shown on the right side, $B \perp\!\!\!\perp A$ and the edge between A and B will be removed first. In the end, we can obtain a more correct graph than the one on the left side.

However, the result of the modified GES is still not perfect; we can see there are redundant edges existing, such as $A \to D$. Therefore, we need the Phase 3 exact search for post-processing. Under the SMR assumption, we can obtain a more sparse graph, where either edge $A \to C$ or edge $A \to D$ will be deleted.

## A4 Proofs

In this section, we provide the proofs of Theorem 2 and Theorem 3 in the main paper.

### A4.1 Proof of Theorem 2

**Proof:** As suggested by the generalized score [32], with proper score functions and search procedures, asymptotically, the resulting Markov equivalence class has the same independence constraints as the data generative distribution.

(i) First of all, we would like to discuss the local consistency of the generalized score.

For the regression problem, one can define the effective dimension of the kernel space and the complexity of the regression function according to [61]. Then under mild conditions, the CV-likelihood score is locally consistent.

**Lemma 6** *Suppose that the sample size of each test set $n_0$ satisfies*

$$n_0 \to \infty, \frac{n_0}{n} \to 0 \text{ as } n \to \infty,$$

*and suppose that the regularization parameter $\lambda$ satisfies*

$$\lambda = O(n^{-\frac{b}{bc+1}}),$$

**Ground-truth Graph:**

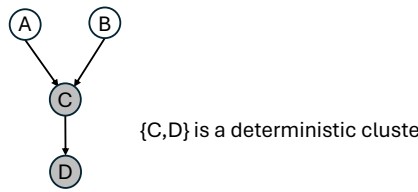

{C,D} is a deterministic cluster.

**Modified GES Forward Phase:** (—→ : added edge)

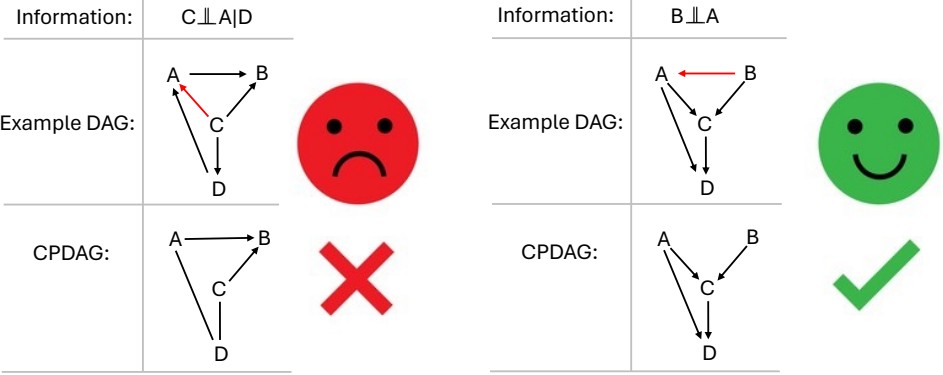

Figure A2: An Example: Original GES vs. Modified GES.

*where $n$ is the total sample size, $b$ is a parameter of the effective dimension of the kernel space with $b > 1$, and $c$ indicates the complexity of the regression function with $1 < c \leq 2$.*

**Lemma 7** *Assume that all conditions given in Lemma 6 hold. With the CV likelihood under the regression framework in RKHS as a score function and with the GES search procedure, it guarantees to find the Markov equivalence class which is consistent to the data generative distribution asymptotically.*

Lemma 7 ensures that, with proper score functions and search procedures, asymptotically, the resulting Markov equivalence class has the same independence constraints as the data generative distribution. For the complete proofs, please refer to the Appendix A5 of paper [32].

(ii) Then, We will provide the proof by contra-positive in both directions based on the consistency of the generalized score as shown above.

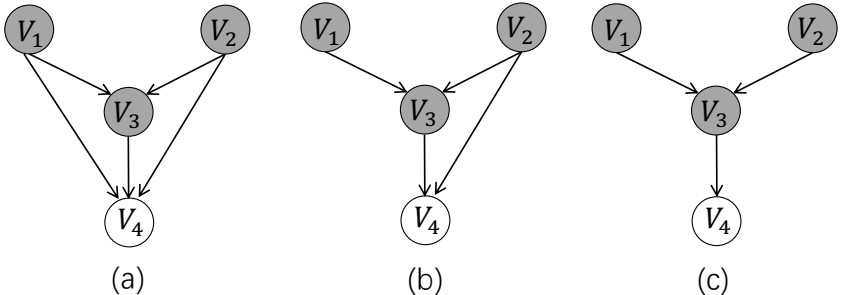

Figure A3: An example graph with determinisic relation where $V_3 = f(V_1, V_2)$. (a) A non-deterministic variable $V_4$ connects to $\{V_1, V_2, V_3\}$. (b) A non-deterministic variable $V_4$ connects to $\{V_2, V_3\}$. (c) A non-deterministic variable $V_4$ connects to $\{V_3\}$. Here among the three graphs, only the graph (c) can be partially identified.

1) "If" direction: Suppose that an exact score-based search asymptotically outputs a DAG $\mathcal{H}$ (having the highest generalized score) that does not belong to the MEC of the true DAG $\mathcal{G}$. Since the generalized score is known to be consistent, $(\mathcal{H}, \mathbb{P})$ must satisfy the Markov assumption because otherwise, its generalized score is lower than that of the true DAG $\mathcal{G}$ and exact search would not have output $\mathcal{H}$. By assumption, the generalized score of $\mathcal{H}$ is higher than that of $\mathcal{G}$, which, by the consistency of generalized, implies that $|\mathcal{H}| \leq |\mathcal{G}|$, and therefore, $(\mathcal{G}, \mathbb{P})$ does not satisfy the SMR assumption.

2) "Only if" direction:
Suppose that $(\mathcal{G}, \mathbb{P})$ does not satisfy the SMR assumption. Then there exists a DAG $\mathcal{H}$ not in the MEC of $\mathcal{G}$ such that $|\mathcal{H}| \leq |\mathcal{G}|$, and $(\mathcal{H}, \mathbb{P})$ satisfies the Markov assumption. Without loss of generality, we choose $\mathcal{H}$ with the least number of edges. We first consider the case in which $|\mathcal{H}| < |\mathcal{G}|$. Since both $\mathcal{H}$ and $\mathcal{G}$ satisfy the Markov assumption, by the consistency of generalized, the generalized score of $\mathcal{H}$ is higher than that of $\mathcal{G}$, which implies that exact score-based search will not output any DAG from the MEC of $\mathcal{G}$. For the case with $|\mathcal{H}| = |\mathcal{G}|$, since they are both Markov with distribution $\mathbb{P}$, they have the same generalized score. Therefore, an exact search will output a DAG that belongs to the MEC of either $\mathcal{H}$ or $\mathcal{G}$ and is not guaranteed to output a DAG from the MEC of the true DAG $\mathcal{G}$.

Proof ends.

### A4.2  Proof of Theorem 3

**Proof:**

First, we will explain why we need the three assumptions listed. Secondly, we will explain why we need to have constraint on $|\mathrm{PA}_i| < |\mathrm{MinDC}| - 1$. Thirdly, we will give the complete proof based on the conditions.

(i) Why do we need the listed three assumptions?

As mentioned in our main paper, there are three phases of our proposed DGES. During the second phase, we need to run GES. To ensure the accuracy of output (particularly on the NDC part), we need the assumptions of Markov and non-deterministic faithfulness (See Assumptions 1 and 3). Then in the third phase, we need to perform the exact search exclusively on the EDC, where the Sparsest Markov Representation (SMR) assumption (See Assumption 2) will be needed.

(ii) Why do we assume $|\mathrm{PA}_i| < |\mathrm{MinDC}| - 1$?

As for why we need to condition on $|\mathrm{PA}_i| < |\mathrm{MinDC}| - 1$, we can start with explaining why $|\mathrm{PA}_i| = |\mathrm{MinDC}|$ and $|\mathrm{PA}_i| = |\mathrm{MinDC}| - 1$ will fail the provided identifiability.

Let's take an example with four variables, where three of them are deterministically related, as shown in Figure A3. Here among the three graphs, only the graph (c) can be partially identified, and the graph (a) and (b) cannot achieve partial identifiability.

We further assume a linear functional causal model, then we can formulate the deterministic relationship as

$$aV_1 + bV_2 + cV_3 = 0, \tag{10}$$

where $a, b, c$ are any linear coefficients. Based on the above formulation, the causal equation of variable $V_4$ in Figure A3(a) can be represented as

$$
\begin{aligned}
V_4 &= dV_1 + eV_2 + fV_3 + \epsilon \\
&= dV_1 + eV_2 + f\frac{1}{c}(aV_1 + bV_2) + \epsilon \\
&= dV_1 + e\frac{1}{b}(aV_1 + cV_3) + fV_3 + \epsilon \\
&= d\frac{1}{a}(bV_2 + cV_3) + eV_2 + fV_3 + \epsilon,
\end{aligned} \tag{11}
$$

where $\epsilon$ is the random noise injected into $V_4$. Clearly, the above four equations are all valid, in other words, $V_4$ can be possibly represented by different sets of variables, meaning that this case is not guaranteed to be identified.

Regarding the variable $V_4$ in Figure A3(b), the causal equation can be represented as

$$
\begin{aligned}
V_4 &= eV_2 + fV_3 + \epsilon \\
&= eV_2 + f\frac{1}{c}(aV_1 + bV_2) + \epsilon \\
&= e\frac{1}{b}(aV_1 + cV_3) + fV_3 + \epsilon.
\end{aligned} \tag{12}
$$

Again, the above three equations are all valid, in other words, $V_4$ can be possibly represented by different sets of variables, meaning that this case is also not guaranteed to be identified.

However, in Figure A3(c), things are different. The causal equation of variable $V_4$ can be represented as

$$
\begin{aligned}
V_4 &= fV_3 + \epsilon \\
&= f\frac{1}{c}(aV_1 + bV_2) + \epsilon.
\end{aligned} \tag{13}
$$

When the SMR assumption is satisfied, we can identify the only one case, which is $V_3 \rightarrow V_4$.

Now, we extend the three-variable case to the general linear case where there is a MinDC with the cardinality $|\operatorname{MinDC}|$. And we can easily conclude the true conditions to be: $|\operatorname{PA}_i| < |\operatorname{MinDC}| - 1$.

Furthermore, we extend the linear to the nonlinear case, where we can also conclude that the listed conditions ensure partial identifiability.

(iii) The complete proof:

Part I:

Benefiting from the local consistency of BIC score (See Lemma 7 of paper [10]) and generalized score (See Proposition 1 of paper [32]), *the NDC part is guaranteed to find the true Markov equivalence class which is consistent to the data generative distribution asymptotically.*

The proof contains two parts: the forward phase and the backward phase of GES. In the forward phase, the resulting equivalence class $\mathcal{E}_f$ contains underlying distribution $\mathbb{P}$; i.e., all independence constraints holding in $\mathcal{E}_f$ hold in $\mathbb{P}$. It has been proved by making use of local consistency of score functions in [10]. Here we focus on showing that the backward phase is guaranteed to find a perfect map of $\mathbb{P}$.

- Let $\mathcal{E}_b$ denote the equivalence class resulting from the backward phase of GES, and let $\mathcal{E}^*$ be the perfect map of $\mathbb{P}$; i.e., all independence constraints in $\mathcal{E}^*$ are in $\mathbb{P}$, and vice versa. Here we aim to show that as the sample size $n \to \infty$, $\mathcal{E}_b = \mathcal{E}^*$.

  1) First, we show that the equivalence class $\mathcal{E}$ results from each step in the backward phase contains $\mathbb{P}$. Consider a move from $\mathcal{E}$ to $\mathcal{E}^-(\mathcal{E})$ by applying Delete$(X_i, X_j, \mathbf{H})$ (see the definition in [10]), where $\mathcal{E}$ contains $\mathbb{P}$ and $\mathcal{E}^-(\mathcal{E})$ does not contain $\mathbb{P}$. Let $\mathcal{G} \in \mathcal{E}$ and $\mathcal{G}' \in \mathcal{E}^-(\mathcal{E})$ with the difference in $X_i \to X_j$. From the fact that the score functions are locally consistent, the local score change $\Delta S < 0$, so $S(\mathcal{G}; D) > S(\mathcal{G}'; D)$. The attempted move from $\mathcal{E}$ to $\mathcal{E}^-(\mathcal{E})$ will be rejected.

  2) Second, we show that the backward phase will not terminate with some suboptimal equivalence class $\mathcal{E}$; that is, there are no independence constraints which containing in $\mathbb{P}$ are not in $\mathcal{E}$. Suppose that the backward phase terminates with some suboptimal equivalence class $\mathcal{E}$, and there is one more edge $X_i \to X_j$ or $X_i - X_j$ in $\mathcal{E}$ than in $\mathcal{E}^*$. According to local consistency, and the calculation of local score change with Delete operator, $\Delta S$ from $\mathcal{E}$ to $\mathcal{E}^*$ is positive; that is, the score of $\mathcal{E}^*$ is larger than that of $\mathcal{E}$. Hence it will move to $\mathcal{E}^*$. It contradicts with the assumption that the backward phase terminates with some suboptimal equivalence class. Therefore, the resulting equivalence class in the backward phase is a perfect map of $\mathbb{P}$.

Part II:

However, the BS part will not be guaranteed to find the true Markov equivalence class so far by GES. Due to the deterministic relations, more dependent edges will be added during the forward phase, e.g., $\{V_1 \to V_6\}$ in Figure 2(b) and $\{A \to D\}$ in Figure A2. However, during the backward phase, all "constant independencies" (i.e., $V_i \perp\!\!\!\perp V_j | S$ with $S \mapsto V_i$ or $S \mapsto V_j$) are ignored indifferently, e.g., $V_1 \perp\!\!\!\perp V_6 | V_2, V_3, V_4$ in Figure 2(b) and $C \perp\!\!\!\perp A | D$ in Figure A2. In the end, the edges $V_1 \to V_6$ and $C \to A$ will be kept.

**Lemma 8 (Sparsity [10])** *Let $\mathcal{G}$ and $\mathcal{H}$ be any two DAGs that contain the generative distribution and for which $\mathcal{G}$ has fewer parameters than $\mathcal{H}$, and let $S$ be any consistent (DAG) scoring criterion. If all DAGs in an equivalence class have the same number of parameters, then for every $\mathcal{G}' \approx \mathcal{G}$ and for every $\mathcal{H}' \approx \mathcal{H}$, $\mathbb{S}(\mathcal{G}'; D) > \mathbb{S}(\mathcal{H}'; D)$.*

Fortunately, we have Phase 3 exact search as post-processing. Under the SMR assumption, we perform the exact search exclusively on the DC and their neighbors. Benefiting from the Lemma 8, a more sparse graph with smaller BS will be selected out of all possible sets. For example, $\{V_3 \to V_6, V_4 \to V_6\}$ will be favoured over $\{V_1 \to V_6, V_2 \to V_6, V_4 \to V_6\}$ in Figure 2, and $\{A \to C\}$ will be favoured over $\{A \to C, A \to D\}$ in Figure A2.

Given the condition $|\operatorname{PA}_i| < |\operatorname{MinDC}| - 1$, the sparsest graph is unique. Therefore, we can identify the BS in such a scenario, e.g., Figure 2. However, when the condition is violated, e.g., Figure A2, we can not uniquely obtain the BS, because both $\{A \to C\}$ and $\{A \to D\}$ can be acceptable BS after executing Phase 3 exact search.

In summary, when the two conditions are satisfied by our DGES, the BS and NDC parts of the causal graph $\mathcal{G}$ are guaranteed to find their true Markov equivalent class, which is consistent with the data generative distribution asymptotically.

Proof ends.

## A5 More Details about the Simulated Experiments

### A5.1 Implementation Details

We provide the implementation details of our method and other baseline methods for synthetic datasets.

**Datasets.** The true DAGs are simulated using the Erdös–Rényi model [33] with the number of edges equal to the number of variables. The data is generated according to the functional causal model $V_i = \sum_{V_j \in \operatorname{PA}_i} b_{ij} f_i(V_j) + \epsilon_i$, where $V_j \in \operatorname{PA}_i$ is the $j$-th direct cause of $V_i$, $\epsilon_i$ is the random noise related to variable $V_i$, and $f_i$ is causal function. For deterministic variables, the noise term is removed,

then the model becomes $V_i = \sum_{V_j \in \text{PA}_i} b_{ij} f_i(V_j)$. We evaluate both linear and nonlinear models. For the linear Gaussian model, we let $f_i(V_j) = V_j$ and $\epsilon_i$ follow Gaussian distribution whose mean is zero and variance is uniformly sampled from $\mathcal{U}(1, 2)$.

For the general nonlinear model, we try two different types. One is by mixed functions, where $f_i$ is randomly chosen from linear, square, sinc, and tanh functions, and $\epsilon_i$ is sampled from uniform distribution $\mathcal{U}(-0.5, 0.5)$ or Gaussian distribution $\mathcal{N}(0, 1)$. The other is generated by MLP, where we consider two hidden layers and each hidden layer has 100 hidden dimensions. We use Sigmiod as the activation function. All the weights are randomly generated from the uniform distribution $\mathcal{U}(0.5, 2)$. For each setting, we also run 10 different random seeds and report the mean and standard deviation.

**Hyperparameters.** During the first phase, when we aim to detect the DC and MinDCs and check whether a variable can be deterministically represented by some others, we set that if the term $\|\Sigma_u\|_{HS}^2 < 1e{-}3$, although theoretically the value should exactly be zero. Meanwhile, the regularization parameter for the kernel ridge regression is set to $1e{-}10$. The second phase of our method is to run modified GES, and the setting is by default. The penalty parameter for controlling the sparsity is set to 1. The exact search in the third phase we incorporate is the A*. We run our method and the other baseline methods in Ubuntu 20.04 LTS 64-bit System with Intel(R) Xeon(R) Silver 4214 2.20GHz $\times$64 CPU. s

**Baselines and Evaluations.** We compare our DGES with other baselines, including DPC [22], GES [10], and A* [15]. We compare the MEC of the output by all methods. For each method, we consider the structural Hamming distance (SHD), the $F_1$ score, the precision, the recall, and the computational time as evaluation criteria. Note that we only evaluate the BS part which we can identify in the graph under mild assumptions. We conduct the experiments on varying number of variables, varying number of samples, and some other hyperparameter studies. For linear model, we evaluate variable $d \in \{8, 10, 12, 14, 16\}$ while fixing sample size $n = 500$, and evaluate sample $n \in \{100, 250, 500, 1000, 2000\}$ while fixing variable $d = 8$. For nonlinear model, we evaluate variable $d \in \{6, 7, 8, 9, 10\}$ while fixing sample size $n = 100$, and evaluate sample $n \in \{50, 100, 150, 200, 250\}$ while fixing variable $d = 6$. We run 10 instances with different random seeds and report the means and standard deviations.

Furthermore, we provide more implementation details for the baseline methods.

- DPC [22]: The method is an extension for traditional PC algorithm [7], the key idea is that: every time when we do the conditional independence test, we aim to remove the potential deterministic variables from the conditioning set so that the faithfulness will not be violated. Here we follow the paper, and use the covariance to measure the closeness of two variables. If the covariance between two variables are greater than 0.9, we then remove the variable from the conditioning set in conditional independence test. Meanwhile, for linear Gaussian model, we choose FisherZ test, while for nonlinear model we choose kernel-based test [44], and the significance level is set to $\alpha = 0.05$ by default. We implement this method based on the Causal-learn package `https://github.com/py-why/causal-learn` [62].

- GES [10]: This method is a classical score-based method with greedy search. Our implementation is based on the code from `https://github.com/juangamella/ges`. For linear Gaussian model, we use BIC score. And for general nonlinear model, we use generalized score with cross-validation likelihood [32]. The penalty parameter for controlling the sparsity is set to 1.

- A* [15]: A* is one of the classical exact score-based methods. Actually, there are some heuristic algorithms proposed to accelerate the search procedure. Considering in our scenarios, we do not utilize any heuristic tricks for the experiments in order to ensure the accuracy of solutions. Our experiments are based on the implementations on the Causal-learn package `https://github.com/py-why/causal-learn` [62].

## A5.2 Evaluation on Two MinDCs

Figure 3 in the main paper presents the simulated results focused on graphs containing just a single deterministic constraint (DC). In contrast, Figure A4 in the Appendix offers insights into scenarios involving two DCs, even allowing for the possibility of overlapping variables. An evident trend

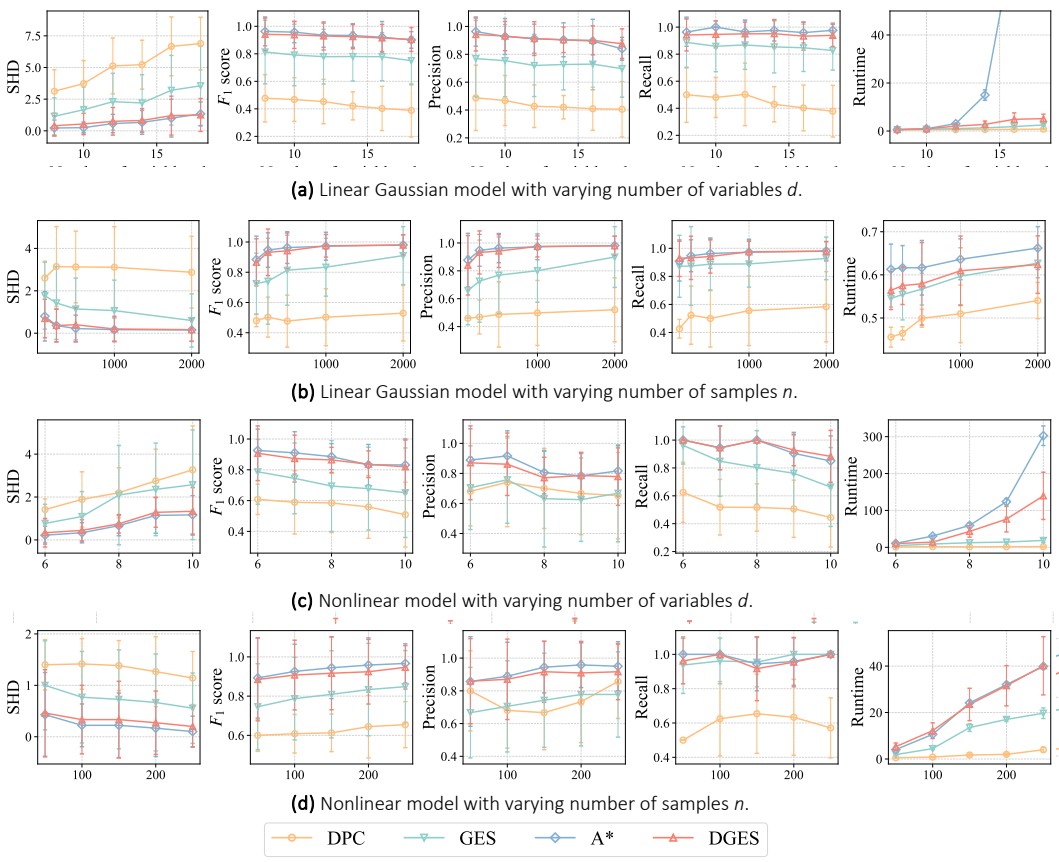

**(a)** Linear Gaussian model with varying number of variables $d$.

**(b)** Linear Gaussian model with varying number of samples $n$.

**(c)** Nonlinear model with varying number of variables $d$.

**(d)** Nonlinear model with varying number of samples $n$.

DPC — GES — A* — DGES

Figure A4: Results on the simulated datasets with two MinDCs. We evaluate different functional causal models on varying number of variables and samples, respectively. For each setting, we consider SHD ($\downarrow$), $F_1$ score ($\uparrow$), precision ($\uparrow$), recall ($\uparrow$) and runtime ($\downarrow$) as evaluation criteria.

emerges: as the system incorporates more deterministic variables, the runtime of our proposed DGES inevitably escalates. This phenomenon can be attributed to the increased number of deterministic variables demanding detection and inclusion in Phase 3, where an exact search is performed.

It is worth noting that as the number of variables in the system increases, the runtime of A* experiences a rapid surge. In stark contrast, DGES exhibits a more stable increase in runtime, demonstrating its efficiency and suitability for both linear and nonlinear models.

The outcomes gleaned from these experiments collectively indicate that DGES exhibits competitive performance compared to established baselines. Notably, the exact method A* and our proposed DGES consistently outperform other baseline methods like Greedy Equivalence Search (GES) and PC, across a spectrum of evaluation criteria and diverse settings. It is intriguing to note that in deterministic systems, the score-based method GES consistently outperforms the constraint-based method DPC. This observation suggests that score-based approaches maintain a comprehensive perspective on causal discovery, which appears to be less susceptible to the challenges posed by deterministic relationships, unlike constraint-based methods.

### A5.3 Evaluation on Non-deterministic Scenario

We also conducted the experiments in a standard setting, where there is no deterministic relation at all. We consider the linear Gaussian model with a varying number of variables. We evaluate the SHD, the $F_1$ score, the precision, the recall, and the runtime. We run 10 instances with different random seeds and report the means and standard deviations.

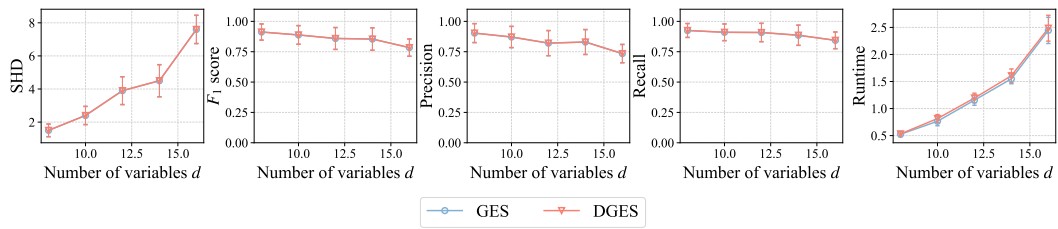

Figure A5: Results of non-deterministic scenarios on linear Gaussian model.

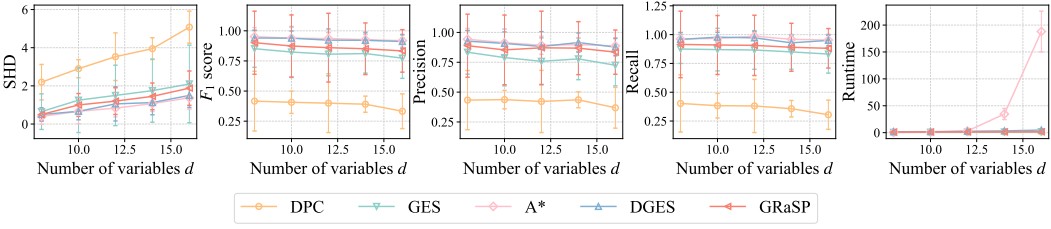

Figure A6: Results of GRaSP [31] on linear Gaussian model.

The results have been shown in Figure A5. According to the results, we can see that GES and our proposed DGES method present the same performance regarding the SHD, the $F_1$ score, the precision, and the recall. However, the runtime of DGES is a bit more than GES, because DGES runs 2 phases. It is understandable that when there is no deterministic relation, DGES will be reduced to GES. In Phase 1, DGES will not find any deterministic clusters, then it will terminate and output the result of GES in Phase 2.

### A5.4 Evaluation on Relaxed Exact Search

GRaSP [31] is a greedy relaxation of the sparsest permutation algorithm. We follow the same setting as mentioned in Section 5. Here we consider the linear Gaussian model with a varying number of variables, and within the generated dataset there is one MinDC. We evaluate the SHD, the $F_1$ score, the precision, the recall, and the runtime. In this case, we evaluate based on only the BS part.

The results have been shown in Figure A6. According to the results, we can see that: in general, A* and our proposed DGES still outperform other baselines. GRaSP performs slightly better than GES regarding the SHD, the $F_1$ score, the precision, and the recall. However, according to our data record, the runtime of GRaSP is a bit more than GES.

## A6 More Details about the Real-world Experiments

Due to the comparative poor performance of DPC and the expensive computation of exact search such as A*, here for the large real dataset, we mainly compare our DGES with GES, in both linear (using BIC score) and nonlinear (generalized score) settings.

### A6.1 Results of GES with BIC Score

In this case, we run GES with BIC score, assuming the model is following linear Gaussian. The causal graph result is given in Figure A7. And in this graph, we can clearly see some deterministic variables are reasonably connected, such as BMI → weight → height.

### A6.2 Results of DGES with BIC Score

We run our proposed DGES with BIC score. The first phase is to identify all the MinDCs. Here, we can detect some MinDCs, such as: {BMI, weight, height.}, {$k_{el}$, $V_d$, Clearance}. The final result is given in Figure A8.

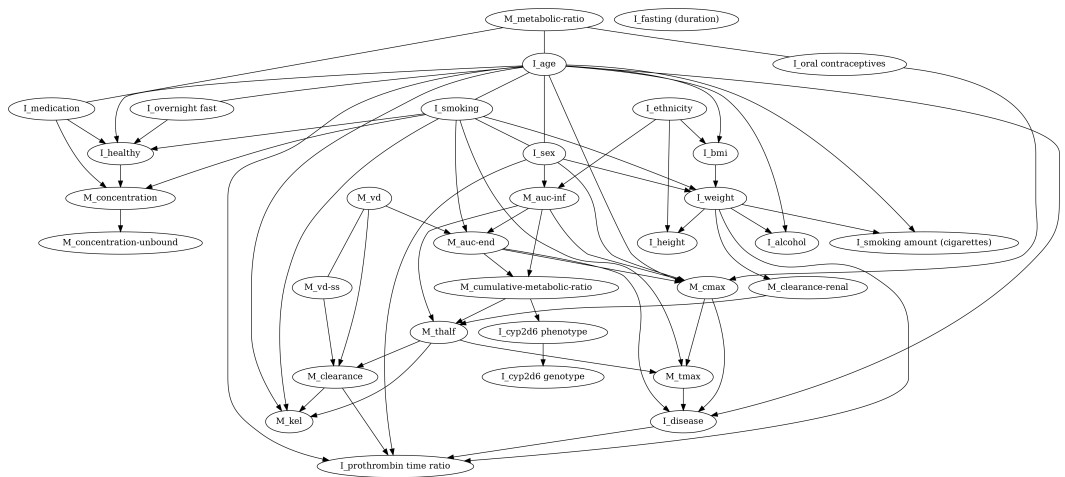

Figure A7: Results of real-world dataset with deterministic relations by GES with BIC Score.

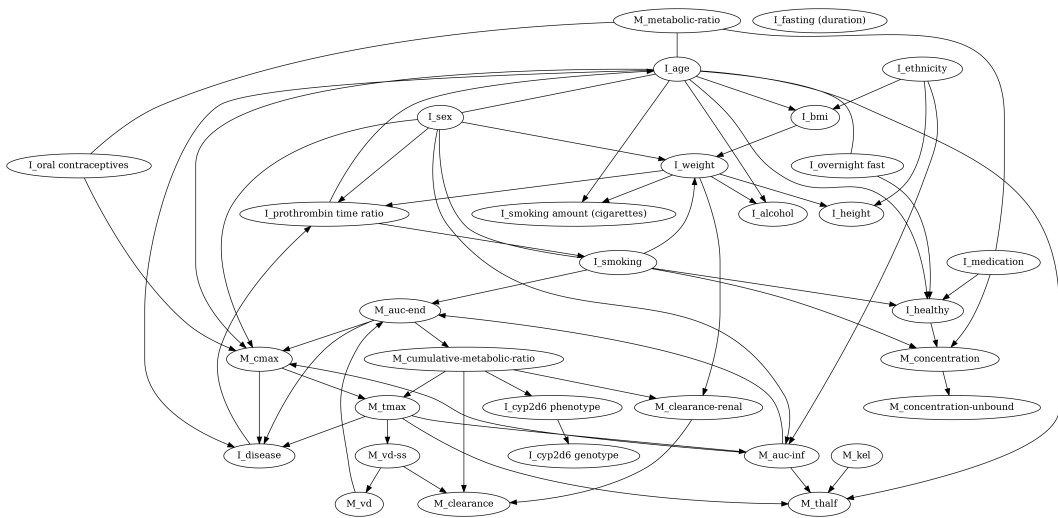

Figure A8: Results of real-world dataset with deterministic relations by DGES with BIC Score.

### A6.3 Results of GES with Generalized Score

BIC score assumes a linear Gaussian model; here, using a generalized score can present general functional models. The GES result with generalized score is given in Figure A9. In this graph, we can still see some deterministic variables are reasonably connected, such as BMI → weight → height.

### A6.4 Analysis of DGES with Generalized Score

This graph is presented in Figure 4 in the main paper. In phase 1, we can detect the following MinDCs: {BMI, weight, height}, $\{k_{el}, T_{half}\}$, $\{k_{el}, V_d, \text{Clearance}\}$, which are all correct.

Specifically, the ground-truth functions are

$$\text{BMI} = \frac{\text{weight}}{\text{height}^2}, \tag{14}$$

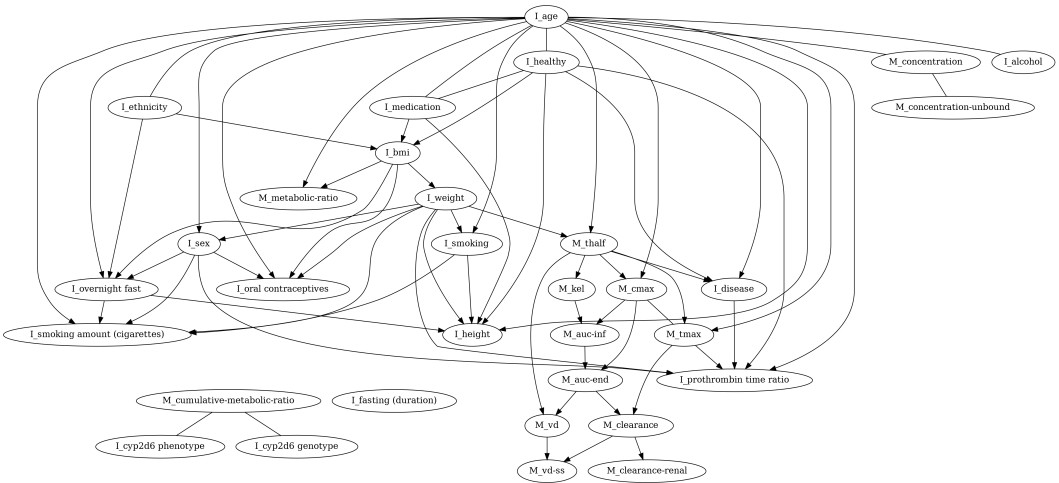

Figure A9: Results of real-world dataset with deterministic relations by GES with Generalized Score.

$$k_{el} = \frac{ln2}{T_{half}}, \tag{15}$$

$$k_{el} = \frac{\text{Clearance}}{V_d}. \tag{16}$$

Take this MinDC {height, weight, BMI} as an example for further analysis. As we all know, BMI is defined as the body weight divided by the square of the body height, which composes a deterministic relation among the three variables. By applying the GES method, the three variables are connected like a cluster comprising a MinDC. At the same time, many other non-deterministic variables still connect with at least one of the three deterministic variables, as usual. Imagine if we use a constraint-based method to deal with it, there should be no edge connecting from the three variables to any others. Furthermore, according to our common knowledge, the true arrows should be {weight→BMI, height→BMI, height→weight}, but now our graph just presents {weight→BMI, weight→height}. As discussed in Appendix A1 (Q1), because currently our method cannot identify the skeleton and directions in a MinDC without further assumptions, and in this case, there are common causes behind BMI and height, which are Healthy and Medication.

Compared with the GES result with the generalized score, we can see some edges are corrected in the DGES result. For example, the MinDC $\{k_{el}, V_d, \text{Clearance}\}$ is clustered together.

Compared with the DGES result with BIC score, we can see more reasonable edges existing in the nonlinear DGES result with the generalized score, for example, {age − medication, healthy → disease, healthy − BMI}.

