# OpenReview forum: "On Causal Discovery in the Presence of Deterministic Relations"
_NeurIPS.cc/2024/Conference — NeurIPS 2024 poster_

### Official Review · Reviewer_Wfp4 · 2024-07-09

**Soundness:** 3
**Presentation:** 3
**Contribution:** 2
**Rating:** 6
**Confidence:** 3

**Summary:**

The paper delves into the challenges of causal discovery from observational data, with a particular focus on deterministic relationships often found in real-world scenarios.
Firstly, the paper demonstrates that exact score-based methods can effectively handle deterministic relationships under mild assumptions. Building on this, it introduces a novel framework called Determinism-aware Greedy Equivalent Search (DGES), which enhances the efficiency and scalability of dealing with deterministic relations and accommodating both continuous and discrete data types.
DGES operates through three key phases: detecting minimal deterministic clusters (MinDCs) in the data, running a modified version of Greedy Equivalent Search (GES) to create an initial causal graph with added constraints for deterministic relations, and performing an exact search on these deterministic clusters and their neighbors to refine the graph and ensure sparsity.
Additionally, the paper establishes partial identifiability conditions for DGES under general functional models and it provides extensive experiments on simulated and real-world datasets to validate the practical efficacy of DGES.

**Strengths:**

* The paper demonstrates that exact score-based methods can effectively handle deterministic relationships under mild assumptions.

* A novel framework called DGES is introduced, which enhances the efficiency and scalability of handling deterministic relations.

* The paper provides conditions under which DGES can achieve partial identifiability for general functional models.

* The theoretical findings and efficacy of DGES are validated through extensive experiments on both simulated and real-world datasets.

**Weaknesses:**

* The algorithm presented in this paper guarantees a partial identifiability of the CPDAG. Which means it is even less informative than a partially oriented DAG.

* The paper's contribution appears somewhat limited, as it addresses the low computational efficiency and poor scalability of exact methods by compromising on general identifiability.

* The paper assumes causal sufficiency which is almost never satisfied in practice.

* Limited experimental results (see questions below)

**Questions:**

* DGES sacrifices general identifiability for improved computational efficiency and scalability. Could you elaborate on the specific scenarios or types of data where this trade-off might be most problematic?

* Can you at least discuss how causal sufficiency can be relaxed? Would it be possible to imagine a combination between DGES and FCI (as it was done for GES and FCI)?

* DGES is designed to accommodate both linear and nonlinear relationships, various data distributions, and both continuous and discrete data. Are there any specific types of data or relationships where DGES performs exceptionally well or poorly?

* The success of exact score-based methods in your approach relies on the SMR assumption. Can you provide more details about this assumption and its practical implications? Do you know if it is satisfied in the real-world dataset?

**Limitations:**

The paper acknowledges certain limitations, such as difficulties in identifying the skeleton and directions in the DC part with overlapping deterministic variables, and the computational expense of Phase 3 when dealing with numerous MinDCs. However, it notes that these searches can often be executed simultaneously.

---

> ### Author Rebuttal · Authors · 2024-08-07
>
> We sincerely appreciate the reviewer’s time and constructive suggestions. With the help of such valuable feedback, we believe that our manuscript could be improved significantly. Please find the point-by-point responses below.
>
>
> **Q1**: "The algorithm presented in this paper guarantees a partial identifiability of the CPDAG. Which means it is even less informative than a partially oriented DAG."
>
> **A1**: Thanks a lot for sharing your concerns. We have discussed the reasons why our method cannot identify the DC part in Appendix A1(Q/A1): to achieve this goal, we usually need strong assumptions on the underlying functional causal model, i.e., linear non-Gaussian model [29]. However, those assumptions are aligned with our goal of a general method.
>
> Fortunately, in some cases, our DGES can still fully identify the DC part up to their CPDAG, such as v-structure, as shown in Figure A1(a/b).
>
> **Q2**: "The paper's contribution appears somewhat limited, as it addresses the low computational efficiency and poor scalability of exact methods by compromising on general identifiability."
>
> **A2**: Thank you so much for pointing it out. As you mentioned, we proposed DGES. Besides that, we want to emphasize two more major contributions. First, we find that exact score-based methods can naturally be used to address the issues of deterministic relations; Importantly, we support our claim with theoretical analysis as presented in Theorem 2.
>
> Secondly, we also provide partial identifiability conditions for our proposed DGES, strengthening our method with theoretical guarantee in Theorem 3. In the appendix A1, we also discussed some cases where we can further identify the DC up to the MEC.
>
> **Q3**: "The paper assumes causal sufficiency which is almost never satisfied in practice. Can you at least discuss how causal sufficiency can be relaxed? Would it be possible to imagine a combination between DGES and FCI (as it was done for GES and FCI)?"
>
> **A3**: We appreciate your insightful question. In fact, causal sufficiency assumption is commonly used in causal discovery, even in some of the most classical methods such as PC [20] and GES [26]. In any case, we agree with you that latent variable or confounder is a significant issues to consider.
>
> To relax causal sufficiency, we can incorporate our DGES framework to those methods which can handle latent variables. Since FCI is constraint-based method relying on conditional independent test while our DGES framework is score-based, it can be challenging to directly combine them.
>
> However, thanks to the recent progress in score-based causal discovery with latent variables, such as SALAD [58], it is absolutely possible to combine our DGES with it, so that we can achieve our goal, i.e., causal discovery in the presence of both deterministic relations and latent variables. As far as we known, SALAD is a exact score-based method assuming linear functional model. In this case, we may incorporate our modification on BIC score as shown in Eq.(3) to SALAD to achieve our goal.
>
>
> **Q4**: "Limited experimental results (see questions below)"
>
> **A4**: We appreciate your constructive comments. We have added one more experiments on a new real dataset. Please check the extra PDF file for more details.
>
> **Q5**: "DGES sacrifices general identifiability for improved computational efficiency and scalability. Could you elaborate on the specific scenarios or types of data where this trade-off might be most problematic?"
>
> **A5**: Thanks a lot for your insightful question. When there are overlapping deterministic relations, our DGES indeed cannot identify the skeleton and directions in DC. In this case, the identifiability of DC part comes to the worst situation. However, there are still some cases, such as v-structure, where our DGES can still fully identify the DC part up to their CPDAG. More discussions can be found in Appendix A1(Q/A1) and Figure A1.
>
> **Q7**: "DGES is designed to accommodate both linear and nonlinear relationships, various data distributions, and both continuous and discrete data. Are there any specific types of data or relationships where DGES performs exceptionally well or poorly?"
>
> **A7**: Thank you so much for your interesting question. According to our experimental results, we found that in linear Gaussian model, our DGES performs quite well, where the $F_1$ score can achieve nearly 95% across different number of variables ranging from 8 to 16 variables. Contrastingly, in the general nonlinear model DGES only achieve nearly 88% $F_1$ score.
>
> **Q8**: "The success of exact score-based methods in your approach relies on the SMR assumption. Can you provide more details about this assumption and its practical implications? Do you know if it is satisfied in the real-world dataset?"
>
> **A8**: Thanks a lot for your constructive question.
>
> More details: The SMR assumption demonstrates that the true DAG G* is the sparsest DAG satisfying the Markov property. The term 'sparsest' refers to the minimal number of edges in the graphical model. Without additional information, the SMR assumption is a necessary condition for any algorithm that uses the CI relations to infer the graph G [21].
>
> Practical implications: The SMR assumption helps in simplifying our causal models by making constraints on the number of edges, which can use the least number of edges to show all the dependence in our data distributions. With no doubt, SMR assumption is much weaker than faithfulness assumption.
>
> Satisfaction in read dataset: Yes, SMR is satisfiable in real-world. We can provide two examples here: in financial markets, stock price changes are often primarily influenced by recent prices and trading volumes, while older data have less impact; and in biological Systems: Gene expression and neural activities are typically driven by a few key genes or neurons, with most other variables having limited influence.

---

> ### Author Response · Authors · 2024-08-07
>
> Reference:
>
> [58] Ignavier Ng, et al. "Score-Based Causal Discovery of Latent Variable Causal Models." ICML, 2024.

---

> > ### Comment · Reviewer_Wfp4 · 2024-08-12
> >
> > I thank the authors for their response. I think they have addressed my concerns. Therefore I will increase my score.

---

> > > ### Author Response · Authors · 2024-08-12
> > > **Thank you so much for checking our responses and increasing your score**
> > >
> > > We are so glad that our responses are helpful to address your concerns. Thank you very much for your constructive feedback and valuable time!

---

### Official Review · Reviewer_j5ZZ · 2024-07-10

**Soundness:** 3
**Presentation:** 3
**Contribution:** 3
**Rating:** 6
**Confidence:** 3

**Summary:**

This paper addresses the challenge of causal discovery in the presence of deterministic relationships by developing a novel framework called Determinism-aware Greedy Equivalent Search (DGES). DGES improves efficiency and scalability in detecting deterministic relations through a three-phase process and is validated with both simulated and real-world datasets.

**Strengths:**

1. The paper is written well and easy to understand.
2. Proposed method is motivated well with theoretical analysis.
3. Experiments are thorough and cover all theoretical insights.

**Weaknesses:**

1. Even if real-world scenarios frequently encounter deterministic relations, the observed data contains noise (e.g., measurement noise) which is very difficult to control. What is the implication of such scenarios?
2. Results does not show superior performance over baselines. This is the major concern.

**Questions:**

See the weaknesses section

**Limitations:**

Limitations are discussed.

---

> ### Author Rebuttal · Authors · 2024-08-07
>
> We greatly appreciate the reviewer’s time and constructive comments. With the help of such valuable feedback, we believe that our manuscript could be improved significantly. Please find the point-by-point responses below.
>
> **Q1**: "Even if real-world scenarios frequently encounter deterministic relations, the observed data contains noise (e.g., measurement noise) which is very difficult to control. What is the implication of such scenarios?"
>
> **A1**: Thank you very much for your insightful question. We totally agree with you that the observed data may contains some noises such as measurement error. In our implementation, we particularly set a threshold to control such noise. When evaluating whether two variables have deterministic relation, we use regression and see if the covariance of noise term is 0, the 0 noise means there exists deterministic relation, as described in Lemma 4 and Lemma 5 in Appendix. Due to the measurement error and so on, the noise covariance will be non-zero even when there is deterministic relation, therefore, we use a small constant as a threshold for evaluation. As mentioned at Line 782 in Appendix, we set such threshold to be $1e{-}3$. In other words, if the noise covariance after regression is smaller than $1e{-}3$, we believe there is a deterministic relation.
>
>
> **Q2**: "Results does not show superior performance over baselines. This is the major concern."
>
> **A2**: Thanks a lot for sharing your concern. In our experiments, we mainly compare our DGES with three baselines: DPC, GES, and A*. We want to emphasize that the the motivation of DGES is to enhance the efficiency and scalability of using exact search (such as A*) to handle deterministic relations. Therefore, our goal is to approximate the accuracy of DGES to that of A*, while improving the time efficiency.
>
>  - As for the comparison with A*, our DGES achieves comparable performance with A* regarding SHD, $F_1$ score, precision and recall, while the runtime of DGES is significantly shorter than that of A*.
>
> - From the results across different sample size, variable size, and different functional causal forms, we can see that our DGES significantly outperforms DPC and GES.
>
>
> ---
>
> Thank you very much again. We hope our responses could address your concerns. Please let us know if you have further comments. Your advice means a lot to us!

---

> > ### Comment · Reviewer_j5ZZ · 2024-08-12
> >
> > I thank the authors for their response. I've read their response and I will increase my score accordingly.

---

> > > ### Author Response · Authors · 2024-08-12
> > > **Thank you so much for checking the responses and updating your recommendation**
> > >
> > > We sincerely appreciate the reviewer for checking our responses and increasing your score. Thanks again for your constructive comments!

---

### Official Review · Reviewer_Nu91 · 2024-07-12

**Soundness:** 3
**Presentation:** 4
**Contribution:** 3
**Rating:** 8
**Confidence:** 4

**Summary:**

This paper focuses on the problem of deterministic dependencies in causal structure learning.  Many algorithms for causal structure learning assume faithfulness between the conditional independencies present in the data and those implied by the graph, and this assumption can be violated when deterministic dependencies are present.  The authors propose a modified version of GES, DGES, for datasets where determinism is present.  DGES adds additional steps to identify deterministic dependencies between variables, performs modified versions of the GES forward and backward passes, and then does an exact search to orient edges around the deterministic nodes.  The authors perform experiments on synthetic data, as well as a real world dataset, and find that DGES consistently performs well.

**Strengths:**

I really like this paper.  Determinism is present in many real-world datasets, but simply removing variables that participate in deterministic relationships may not be practical or may result in other issues.  DGES is well-described and, while a relatively simple modification to an existing algorithm, the authors provide a strong theoretical basis and detailed algorithmic description, helping it stand as a solid contribution.  The synthetic experimental results are interesting and well-done, comparing across multiple parameter settings and against relevant comparison algorithms.  Overall, i think this is a valuable paper.

**Weaknesses:**

This paper could use some work on the motivation.  In the introduction and Section 2, the authors convincingly show that the PC-algorithm (and other constraint-based algorithms) does not work in the presence of deterministic dependencies.  The authors describe score-based methods in the introduction (though don't say either way if they generally handle determinism) and explain that some exact score-based methods are able to handle determinism just fine.  From this, the take-away seems to be "don't use constraint-based methods; use exact score-based methods."  However, the authors then go on to expand on GES, a non-exact score-based methods, proposing a modified version of it as their solution.  In Section 2.3, the authors then point out that exact score-based methods are computationally inefficient - this is good to know, but would have been helpful to also mention in the introduction.  However, to this point, I'm still left unsure about whether or not GES can handle determinism as-is, which is strange, given that it's the basis of the proposed DGES.  It in't until Section 3.3 that I think I got my answer to (1) - "As demonstrated by Lu et al. [19], GES may get sub-optimal results when the faithfulness assumption is violated".  This is important motivation and should be present in the introduction or, at the very least, Section 2.  Saving it until page 7/9 just leaves the reader unsure about the necessity of the proposed method for far too long.

I also think the explanation provided (i.e., someone else showed that it may be "sub-optimal") is a bit lacking, given that it's partially the foundation of this work. (it's clear from the experimental results that GES does not handle determinism well, but simply saying it "may" be "sub-optimal" is vague and gives no sense of the severity of the issue) Apart from the quoted line in Section 3.3, Section 3.2 says that the authors "add some extra constraints during the forward and backward steps and adjust the score function due to the deterministic relations."  The authors then go on to describe GES and their modifications, but I don't see any discussion about the motivation for those modifications.  Stronger justification would help a lot before getting into the details.

In Theorem 2, you say that exact score-based search works if the SMR assumption is satisfied and also "some mild conditions are satisfied".  "Mild conditions" could be basically anything - at the very least, allude to what type of conditions they are, even just in a footnote.

In line 204, the line "we need to traverse all the possible combination sets of DC" is odd and unclear.  What is "all the possible combination sets of deterministic cluster"?  I thought there was only one DC, so I don't know how we know have combinations of deterministic clusters... I think this is referring to all combinations of variables within the DC??

In Figure 3, I don't see the units for runtime, either in the graph or in the text.  Please add those, unless I'm just missing them somewhere.

The one real-world dataset is weak.  I don't believe there is any ground truth being compared against, correct?  As it stands, I'm not sure what I'm supposed to get out of Figure 4.  It's a very busy graph with a lot of abbreviations, so the takeaway just ends up being "DGES can output a graph".  You then call-out in the text that DGES was able to detect 3 MinDCs, but it's hard to verify this, since they're not highlighted in any way in Figure 4.  If you want to include Figure 4 here, marking the edges that are deterministic vs probabilistic separately would help a lot.  Also, the text makes it sound like DGES does great on this dataset by calling out, for example, that it got the MinDC {height, weight, BMI}.  However, this comes across a bit disingenuous.  We know the ground truth is height -> BMI <- weight.  When I located those variables in Figure 4, however, the structure I actually see is BMI -> weight -> height, which is definitely wrong.  I'd be interested to know how often other methods are able to correctly orient deterministic functions like this one in the data.  Looking at Appendix 6, GES appears to make the same mistake (BMI -> weight -> height).  However, the text in the main paper discussing these results comes across as fully positive about the performance of DGES.  Some acknowledgement that these MinDCs are not perfectly determined (and making that a lot easier to see in Figure 4 by marking the deterministic edges and maybe highlighting these specific clusters that you reference) would help a lot.

These don't affect my score, but there are a number of typos and grammatical issues.  I'd recommend another editing pass or two.  Some examples I noted:
- line 52 - sparest -> sparsest
- line 55 - [The] "d-separation condition is proposed"... (need an article)
- line 63 - missing period after "works"
- line 71 - "...graphs, therefore, we propose..." is a run-on sentence
- line 302 - "As the number of variable increasing" -> "As the number of variables increases"

**Questions:**

Can you explain more about how Assumption 2 functions as an assumption, as opposed to as a definition?  I understand the idea of the "sparsest MEC which satisfies the Markov assumption", so is the idea behind the SMR assumption that we assume the algorithm will return not just the MEC but the sparsest MEC?

Did you try any experiments where you know there are no deterministic variables present?  It would be helpful to know if it's safe to just always use DGES, even if we're not 100% sure that there is determinism.

**Limitations:**

I think for the most part, the authors address the limitations of DGES.  The only pieces I think I'm missing are what the "mild conditions" are for general identifiability of exact search and how DGES performs in situations with no determinism.



Note: The authors adequately addressed my concerns, so I am increasing my score from 7 to 8.

---

> ### Author Rebuttal · Authors · 2024-08-07
>
> We deeply appreciate the reviewer for your time dedicated to reviewing our paper, encouraging words and constructive suggestions. In light of your valuable feedbacks, we have carefully modified the structure and narrative of our manuscript. Please find the responses to all your comments point-by-point below.
>
> -----------------------
>
> **Q1**: "This paper could use some work on the motivation ... This is important motivation and should be present in the introduction or, at the very least, Section 2. Saving it until page 7/9 just leaves the reader unsure about the necessity of the proposed method for far too long."
>
> **A1**: Thank you for your constructive suggestions. In light of your comments, we would like to add a few more sentences at the end of Line 54, to clarify our motivation. Below is the content:
>
> "Under SMR assumption [21], we can use exact score-based methods such as DP [12] and A* [14] to handle deterministic relations. Even though faithfulness is violated, we can still get reliable result as long as SMR is satisfied. However, due to the large search space of possible DAGs, exact score-based methods should be inefficient. GES is an efficient score-based method in a greedy manner. As demonstrated by Lu et al. [19], GES may get sub-optimal results when the faithfulness assumption is violated, e.g., when there are deterministic relations. To tell the difference between sub-optimal and optimal results, we provide an example in Figure 2 where the optimal graph (ground truth graph) is on the left and the sub-optimal result is on the right (graph by GES). Based on the sub-optimal result by GES, we can identify those problematic edges and further correct them. To that end, in this paper, we propose an efficient three-phase method for dealing with determinism."
>
>
> **Q2**: "I also think the explanation provided (i.e., someone else showed that it may be "sub-optimal") is a bit lacking, given that it's partially the foundation of this work. (... simply saying it 'may' be 'sub-optimal' is vague and gives no sense of the severity of the issue)."
>
> **A2**: Thanks a lot for raising this point. Actually, as shown in Figure 2, we have provided an example to explain the difference between the optimal result (ground truth graph) and the sub-optimal result (graph by GES). In this example, GES may get some edges {$V_1, V_2, V_4$} $\rightarrow V_6$. However, the ground truth exhibits a more sparse graph by {$V_3, V_4$} $\rightarrow V_6$. In light of your comments, we will incorporate the statement above into Section 3.3 to make our motivation clearer.
>
>
> **Q3**: "Section 3.2 says that the authors "add some extra constraints during the forward and backward steps and adjust the score function due to the deterministic relations." The authors then go on to describe GES and their modifications, but I don't see any discussion about the motivation for those modifications. Stronger justification would help a lot before getting into the details."
>
> **A3**:  Thank you very much for sharing your concerns. Actually as mentioned in Line 236, we have discussed our motivations with a detailed example in Appendix A3.2 and Figure A2. We totally agree with you that motivations should be discussed before we get into details. Therefore, we will add more contents at the beginning of Section 3.2 to clarify our motivations. The added contents are:
>
> "We modify the standard GES [10] in the forward and backward phases, and adjust the score function due to the deterministic relations. The key modification in the forward and backward phases is that we always regard the relations to be dependent whenever there is deterministic relation. That is to say, we always assume $V_i \not\perp V_j | PA_{j}$, when $PA_{j}$ determines $V_i$ or $V_j$. The motivations are in the following. During the forward phase, we want to preserve as much dependent edges as possible, so that the BS will not be empty due to determinism, as shown in Figure 1(a). During the backward phase, ignoring the dependence in deterministic relations can lead to wrong edges in the NDC part; A motivating example is given in Appendix Figure A2. After the forward and backward phases, we can guarantee that output equivalent class will be Markovian to the ground truth, although some redundant edges may exist. Fortunately, we have Phase 3 exact search as post-processing, which will be introduced in Section 3.3."
>
> **Q4**: "In Theorem 2, you say that exact score-based search works if the SMR assumption is satisfied and also "some mild conditions are satisfied". "Mild conditions" could be basically anything - at the very least, allude to what type of conditions they are, even just in a footnote."
>
> **A4**: "Thank you so much for your constructive advice. Here, mild assumptions are used to ensure that the generalized score is locally consistent. Specifically, these assumptions include constraints on the infinite sample size and the value of the regularization parameter $\lambda$. More details are provided in Appendix Lemma 6, which is adapted from Lemma 2 of paper [32]. In light of your comments, we will add the explanations above to our main paper."

---

> ### Author Response · Authors · 2024-08-07
> **Responses (2/3)**
>
> **Q5**: "In line 204, the line "we need to traverse all the possible combination sets of DC" is odd and unclear. What is "all the possible combination sets of deterministic cluster"? I thought there was only one DC, so I don't know how we know have combinations of deterministic clusters... I think this is referring to all combinations of variables within the DC??"
>
> **A5**: Thanks a lot for pointing out this question. Indeed there was only one DC, however, there could be multiple MinDCs within this DC, because there might be multiple deterministic relations, even with overlapping deterministic variables. Let's consider this overlapping example, where the DC is {$V_1, V_2, V_3, V_4, V_5$}, $\{V_1,V_2\}\mapsto V_3$, and $\{V_2,V_4\}\mapsto V_5$. In this case, once we obtained the DC, we can further detect two MinDCs by iterating all possible combination variable subset of DC. Finally we can get {$V_1, V_2, V_3$} and {$V_2, V_4, V_5$}. We rely on MinDCs to run the modified GES. More details regarding how to get DC and MinDC are given in Appendix A3.1 and Algorithm A1/A2. In light of your comments, we will incorporate this example and explanation above into Section 3.1 to make our descriptions clearer.
>
>
> **Q6**: "In Figure 3, I don't see the units for runtime, either in the graph or in the text. Please add those, unless I'm just missing them somewhere."
>
> **A6**: Thank you very much for pointing out this issue and for your careful reading. All units here are in seconds. We will update this information in our main paper.
>
>
> **Q7**: "The one real-world dataset is weak. I don't believe there is any ground truth being compared against, correct? As it stands, I'm not sure what I'm supposed to get out of Figure 4. It's a very busy graph with a lot of abbreviations, so the takeaway just ends up being "DGES can output a graph". You then call-out in the text that DGES was able to detect 3 MinDCs, but it's hard to verify this, since they're not highlighted in any way in Figure 4. If you want to include Figure 4 here, marking the edges that are deterministic vs probabilistic separately would help a lot."
>
> **A7**: Thanks a lot for sharing your concerns. Let us answer your questions one by one. (1) Yes, the real dataset we use has no ground truth. What we know is based on domain experts' knowledge, such as: BMI = $weight / height^2$, $K_{el} = Clearance / V_d$, and so on. Using this expert knowledge, we evaluate and compare the results of DGES with those of GES to see how it improves. (2) In light of your comments, we will re-draw the causal graph, highlight the MinDC variables, and update all of the graphs for real-world datasets in the revised version.
>
>
> **Q8**: "Also, the text makes it sound like DGES does great on this dataset by calling out, for example, that it got the MinDC {height, weight, BMI}. However, this comes across a bit disingenuous. We know the ground truth is height -> BMI <- weight. When I located those variables in Figure 4, however, the structure I actually see is BMI -> weight -> height, which is definitely wrong. I'd be interested to know how often other methods are able to correctly orient deterministic functions like this one in the data. Looking at Appendix 6, GES appears to make the same mistake (BMI -> weight -> height). However, the text in the main paper discussing these results comes across as fully positive about the performance of DGES. Some acknowledgement that these MinDCs are not perfectly determined (and making that a lot easier to see in Figure 4 by marking the deterministic edges and maybe highlighting these specific clusters that you reference) would help a lot."
>
> **A8**: We appreciate your thoughtful questions. Regarding the structure of MinDC {height, weight, BMI}, the true graph should be a fully connected graph because besides height -> BMI <- weight, height and weight are also dependent. As mentioned in Appendix A1(Q/A1), we did acknowledge that our method cannot perfected identify the skeleton and directions in the DC part. However, as shown in Figure 4, our DGES method can still find the dependence within the MinDC, for example, there are edges between BMI and weight, weight and height. Meanwhile, BMI and height are also dependent due to their common cause 'I_healthy'.
>
> We want to emphasize that our DGES method aims to give a general framework to identify the BS and NDC parts, if we want to further identify the orientations of the MinDC part, further strong assumptions on functional causal form will be needed, e.g., paper [29] assumed linear non-Gaussian model.

---

> ### Author Response · Authors · 2024-08-07
> **Responses (3/3)**
>
> **Q9**: "There are a number of typos and grammatical issues. I'd recommend another editing pass or two."
>
> **A9**: "Thank you for your careful reading and pointing out the typos. We will correct the noted sentences as follows:
>
> - Line 52: "When the sparsest Markov representation (SMR) is satisfied"
>
> - Line 55: "Deterministic relations have been considered in a few works of causal discovery [22-29]. The 'D-separation' condition [7] is proposed..."
> - Line 63: "However, there is no identifiability guarantee in those related works. Moreover..."
> - Line 71: "the exact score-based methods are feasible only for small graphs, and can be inefficient for large graphs. To enhance the efficiency and scalability, we propose a novel framework called DGES."
> - Line 302: "As the number of variables increases.."
>
> Your comments indeed help to improve the quality of our paper. We will update the above sentences and polish our paper thoroughly in light of your comments.
>
>
> **Q10**: "Can you explain more about how Assumption 2 functions as an assumption, as opposed to as a definition? I understand the idea of the "sparsest MEC which satisfies the Markov assumption", so is the idea behind the SMR assumption that we assume the algorithm will return not just the MEC but the sparsest MEC?"
>
> **A10**: Thanks for your interesting question. You are totally correct that we assume the algorithm will return the unique sparsest MEC, not just the MEC. The SMR assumption demonstrates that the true DAG G* is the sparsest DAG satisfying the Markov property. The term 'sparsest' refers to the minimal number of edges in the graphical model. The SMR assumption helps in simplifying our causal models by making constraints on the number of edges, which can use the least number of edges to show all the dependence in our data distributions.
>
>
> **Q11**: "Did you try any experiments where you know there are no deterministic variables present? It would be helpful to know if it's safe to just always use DGES, even if we're not 100% sure that there is determinism."
>
> **A11**: Thank you so much for your thoughtful question. Actually we have conducted the experiment under linear Gaussian model with no deterministic relations at all. The result is in Figure A5 and the analysis is in Appendix A5.3. Basically, the performance of DGES almost aligns with that of GES regarding SHD, $F_1$, Precision and Recall. The runtime of DGES (unit: second) is a little bit higher than that of GES, due to the detection of MinDC in the first phase. Since the DC is detected to be empty, we have no need to further detect MinDCs, therefore phase 1 would still be fast.
>
>
> Should there be any further questions or concerns, please let us know and we stand ready and eager to address them. We highly value your insights and would be more than pleased to provide any additional information or clarification you may require.

---

> > ### Comment · Reviewer_Nu91 · 2024-08-13
> >
> > Thank you for the detailed response!  With the proposed changes, I will update my score.

---

> > > ### Author Response · Authors · 2024-08-13
> > > **Thank you very much for your valuable time and insightful comments**
> > >
> > > We sincerely appreciate the time and effort you invested in carefully evaluating our paper. Your constructive and insightful feedback has greatly enhanced our work. Thank you very much!

---

### Official Review · Reviewer_zDFG · 2024-07-14

**Soundness:** 3
**Presentation:** 4
**Contribution:** 3
**Rating:** 7
**Confidence:** 4

**Summary:**

Summary
In this paper, the authors develop an approach to causal discovery with deterministic casual relations. They adapt the GES algorithm to deal with common faithfulness violations due to spurious conditional independences.

**Strengths:**

Strengths
- The paper is well-written and the ideas are clearly presented
- The proposed method seems like it should be possible to adapt to other causal discovery algorithms
- Assumptions made in causal discovery are generally very strong, and finding approaches to alleviate these is an important topic

**Weaknesses:**

Weaknesses
- It's not quite clear how common or relevant these deterministic relationships are, and how different the resulting output networks really are
- The theoretical results seem like they could be improved (Q6)
- The evaluation on only a single real-world dataset seems rather weak (Q8,9)

**Questions:**

Questions
1. Is there any particular reason why we adapt GES specifically? Could we adapt other score-based methods in the same way?
2. Can you explain how the results you obtain differ from those obtained by other methods to deal with faithfulness violations, e.g. [1]?
3. Is there anything we can do in the low sample, high-dimensional case, where every variable's sample can be written deterministically in terms of other variables?
4. Currently you search over all subsets of DC to find the MinDC a variable belongs too. Is there any way to make this more efficient?
5. Can you explain why in (3) you need to add a small constant to the covariance, but in the corresponding terms in (4) you do not?
6. Theorem 3 seems like it could be improved. For example, if $X_{2i-1} \rightarrow X_{2i}$ for all $i$, but no other edges exist, then we should be able to say more about the independence of different MinDCs from each other?
7. In Figure 3(d), and particularly (b), why is the runtime of DGES almost as high as $A^\star$?
8. In the real-world dataset, are any of the discovered MinDCs interesting? All three of them seem to be rather simple. Are there example networks where the MinDCs would not be predictable by domain experts?
9. On a related note, does the graph obtained by specifically including these deterministic relations qualitatively differ from what we would have found without them? Is BMI a causally relevant variable in the first place, rather than a--possibly rather arbitrary--construct?

References
[1] A Weaker Faithfulness Assumption based on Triple Interactions

**Limitations:**

It's currently not clear how relevant the ability to deal with these deterministic relationships truly is

---

> ### Author Rebuttal · Authors · 2024-08-07
>
> We sincerely appreciate the reviewer’s time and constructive suggestions. With the help of such valuable feedback, we believe that our manuscript could be improved a lot. Please find the point-by-point responses below. Q1-Q9 correspond to the points in "Questions", while Q10-Q12 correspond to the points in “Weaknesses”.
>
> **Q1**: “Is there any particular reason why we adapt GES specifically? Could we adapt other score-based methods in the same way?”
>
> **A1**: Thanks a lot for your interesting questions. (1) Yes. This paper mainly focuses on score-based causal discovery methods, and GES is one of the most typical and well-known methods in this category with theoretical guarantee, that is why we adapt GES specifically. (2) It depends on which type of score-based methods to consider. For example, for those greedy score-based methods such as GIES [53] and GES-mod [54], we can adapt them in the same way. However, for those continuous-optimization score-based methods such as NOTEARS [39], we cannot directly adapt them.
>
> **Q2**: "Can you explain how the results you obtain differ from those obtained by other methods to deal with faithfulness violations, e.g. [55]?"
>
> **A2**: We appreciate you pointing out the relationships between deterministic relations and faithfulness violation. To tell the difference of the two outputs, we can provide a simple example with deterministic relations where our DGES method can work while the 2-Adjacency faithfulness assumption is still violated, and thus paper [55] doesn't work. The example is: $X_2 \leftarrow X_1 \rightarrow X_3$ where $X_1 \mapsto X_2$. DGES can find the edges {$X_2-X_1, X_1-X_3$}. However, for the adjacent pair $X_1$, $X_3$, the MB of $X_1$ cannot form any two variables to render 2-Adjacency faithfulness, as $X_1 \perp X_3 | X_2$.
>
> To sum up, those faithfulness relaxation methods such as [55] work on general faithfulness violation and propose some weaker faithfulness assumptions. They usually focus on certain types of structure, such as canceling path, XOR-type, triangle faithfulness, etc. However, to the best of our knowledge, deterministic relations will break all those relaxed faithfulness assumptions, as the distribution is even not a graphoid. Therefore, we need to develop specific algorithms to handle determinism.
>
> In light of your comments, we will add more discussions in the relate work section, to tell the relationshipa between ours and faithfulness relaxation methods including paper [55].
>
>
> **Q3**: "Is there anything we can do in the low sample, high-dimensional case, where every variable's sample can be written deterministically in terms of other variables?"
>
> **A3**: Thanks for your insightful questions. In the low sample and high-dimensional case, basically all variables belong to the DC, and the NDC and BS will be empty set. In such a case, our DGES will be invalid, because DGES mainly aims to identify the BS and NDC parts.
>
> **Q4**: "Currently you search over all subsets of DC to find the MinDC a variable belongs too. Is there any way to make this more efficient?"
>
> **A4**: Thanks a lot for your thoughtful question. One possible way to make it efficient is to conduct the pruning process to decrease the search space. For example, once we obtain one MinDC, we can directly eliminate all the superset of this MinDC.
>
> **Q5**: "Can you explain why in (3) you need to add a small constant to the covariance, but in the corresponding terms in (4) you do not?"
>
> **A5**: Thank you very much for your careful reading. When dealing with deterministic relations using BIC score in Eq.(3), the estimated variance of the noise term $|\Sigma|$ will get close to 0, then $\log|\Sigma|$ will encounter with arithmetic error because of $\log0$, here we add a small constant to avoid such an issue. However, in Eq.(4), due to the ridge kernel regression with postive regularization parameter $\lambda$, the estimated covariance matrix is already positive-definite, therefore, we do not need extra small positive constant.
>
> **Q6**: "Theorem 3 seems like it could be improved. For example, if $X_{2i-1} \rightarrow X_{2i}$ for all $i$, but no other edges exist, then we should be able to say more about the independence of different MinDCs from each other?"
>
> **A6**: Thank you for your constructive comments. You are totally correct that Theorem 3 can be further improved. Actually we had more discussion on Theorem 3 in Appendix A1, particularly discussing when we can identity the DC part. In Figure A1, we presented the two cases where the whole causal graph can be identified up to their Markov equivalent class (MEC). The first case in Figure 1A(a) is $V_1\rightarrow V_2$ where $V_1$ and $V_2$ have a deterministic relation, which is exactly a simplified version of what you described ($X_{2i-1} \rightarrow X_{2i}$ for all $i$). In this case, our DGES can indeed achieve full identifiability up to true MEC over all DC, NDC and BS parts.
>
> **Q7**: "In Figure 3(d), and particularly (b), why is the runtime of DGES almost as high as A*?"
>
> **A7**: Thanks for your careful reading and raising this concern. In Figure 3(b) and 3(d), we fixed the number of variable in a rather small value ($d=8$ in linear model and $d=6$ in nonlinear model) and evaluated how increasing sample size effected the performance. In a small variable number, A* can perform accurately and efficiently. Meanwhile, we want to emphasize that the time costs of our DGES method include three parts: detect MinDC, run modified GES, and run exact search as postprocessing. Therefore, when summing time costs of all three phases together, particularly in a rather small variable case, the runtime of DGES is comparable with A*.

---

> ### Author Response · Authors · 2024-08-07
> **Responses (2/3)**
>
> **Q8**: "In the real-world dataset, are any of the discovered MinDCs interesting? All three of them seem to be rather simple. Are there example networks where the MinDCs would not be predictable by domain experts?"
>
> **A8**: We appreciate your interesting question. Yes, the discovered DC {$K_{el}$, $V_d$, Clearance, $T_{half}$} is interesting because of the overlapping variable $K_{el}$. By the Phase 1 of our method, we can successfully detect the MinDCs: {$K_{el}$, $V_d$, Clearance}, {$K_{el}$, $T_{half}$}, and {$T_{half}$, $V_d$, Clearance}.
>
> We learn two biology equations from domain experts: $K_{el}$ = $V_d$ / Clearance, and $T_{half}$ = $ln2$ / $K_{el}$. The two equations are consistent with what we have detected. So far, our detected MinDCs in the real-world dataset are all predictable or explainable by domain experts, because all those variables are well-studied.
>
> In fact, the MinDC detection phase of our DGES does NOT require any prior knowledge, which means that it is absolutely possible to discover more unknown or unpredictable deterministic relations, when given a set of unknown or new variables.
>
> **Q9** "On a related note, does the graph obtained by specifically including these deterministic relations qualitatively differ from what we would have found without them? Is BMI a causally relevant variable in the first place, rather than a--possibly rather arbitrary--construct?"
>
> **A9**: Thanks for your insightful question. Regarding the first question, the graphs obtained with or without deterministic relations can be totally different. Let's use an example to see the differences.
>
> - Consider a four-variable graph: {$X_1 \rightarrow X_2 \leftarrow X_3, X_3 \rightarrow X_4$}, where $\{X_1, X_3\}\mapsto X_2$ and {$X_1,X_2,X_3$} makes up a MinDC.
>
> - If we simply remove $X_1$, we will obtain {$X_2 - X_3 - X_4$};
>
> - If we simply remove $X_2$, we will obtain {$X_1 - X_3 - X_4$};
>
> - If we simply remove $X_3$, we will obtain {$X_1 \rightarrow X_4 \leftarrow X_2$};
>
> We can see that the three outputs are equally sparse, however, graphs are different from each other. Without additional information or prior knowledge, it is difficult to decide which variable from MinDC to remove and how to reconstruct the graph by putting the removed variable back.
>
> As for the second question, we believe that BMI is a causally relevant variable. The reasons are: BMI is strongly associated with body fat percentage in the general population; Usually high BMI values are associated with increased risk of various health conditions, such as cardiovascular disease; In Figure 4, our estimated causal graph shows the clear dependence between health condition ("I_health") and BMI value ("I_BMI"), which is reasonable.
>
>
> **Q10**: "It's not quite clear how common or relevant these deterministic relationships are, and how different the resulting output networks really are."
>
> **A10**：We appreciate your constructive comments. Actually deterministic relations are quite common in real-world, particularly in biology science [56, 57]. Take Figure 4 in our main paper as an example, we can see that the variables within one MinDC are connected, meaning their strong relevance and dependence. More analysis about the real-world dataset has been given in Appendix A6. We also compare different output networks. At Line 888, we compare the results by DGES with that by GES; Given the detected MinDC {$K_{el}$, $V_d$, Clearance}, DGES can further refine the BS and MinDC parts by GES. At line 890, we also compare generalized score with BIC score in DGES, we can see more reasonable edges are detected by generalized score, such as {age-medication, healthy-disease, healthy-BMI}. Since the relations are more likely to be nonlinear in real datasets, it is more reasonable to use generalized score in a non-parametric way.
>
> **Q11**: "The theoretical results seem like they could be improved."
>
> **A11**: Thanks for your kind suggestion. Please refer to Q6/A6 for more details about our response.
>
> **Q12**: "The evaluation on only a single real-world dataset seems rather weak (Q8,9)."
>
> **A12**: Thank you so much for your advice. We have conducted the experiments on another real-world dataset. Please check the global "Official Comment" above.
>
> Thank you very much again. We hope our responses could address your concerns. Please let us know if you have further comments. Your advice means a lot to us!

---

> > ### Comment · Reviewer_zDFG · 2024-08-08
> >
> > Thank you for your extensive response, I have only a few follow-up comments/questions.
> >
> > Q8/10: You say that it is absolutely possible to determine more deterministic relationships. However, the deterministic relationships shown here exist because that is how we have defined them. E.g., BMI is not a quantity measured independently of height and weight, it is defined in terms of these two quantities.
> >
> > Q9: First, I am afraid I do not understand your example here. $X_1, X_3$ are unconditionally independent since $X_2$ is a collider, and removing $X_2$ does not condition on it. Therefore the graph upon removing $X_2$ should simply contain the edge $X_3 \to X_4$? Second, your explanation as to why BMI matters is about association. However, BMI itself is unlikely to be causal for health issues, and contains no information that is not already contained in height and weight. What is the point in keeping the variable, when all predictions based on it could already be made based on height and weight?

---

> ### Author Response · Authors · 2024-08-07
> **Responses (3/3)**
>
> References:
>
> [53] A. Hauserand and P.Bühlmann. "Characterization and Greedy Learning of Interventional Markov Equivalence Classes of Directed Acyclic Graphs." Journal of Machine Learning Research, 2012.
>
> [54] J. I. Alonso-Barba, et al. "Scaling Up the Greedy Equivalence Algorithm by Constraining the Search Space of Equivalence Classes." Internat. J. Approx. Reason., 2013.
>
> [55] Alexander Marx, et al. "A Weaker Faithfulness Assumption based on Triple Interactions." UAI, 2021.
>
> [56] Niklas Gericke, et al. "Exploring Relationships among Belief in Genetic Determinism, Denetics Knowledge, and Social Factors." Science & Education, 2017.
>
> [57] Attila Grandpierre, et al. "The Universal Principle of Biology: Determinism, Quantum Physics and Spontaneity." NeuroQuantology, 2014.

---

> ### Author Response · Authors · 2024-08-08
> **Response to Reviewer zDFG**
>
> We really appreciate you taking the time to share your valuable comments and questions so promptly. Our responses to these questions are given below.
>
> First of all, we apologize for the typo in A9. The four-variable example we had in mind was: {$X_1 \rightarrow X_3 \leftarrow X_2, X_3 \rightarrow X_4$}, where {$X_1,X_2$} $\mapsto X_3$ (deterministic relation) and $X_3 \rightarrow X_4$ is a normal edge with random noise.
>
>   - If we simply remove $X_1$ and run PC on the remainder, we will obtain {$X_2 - X_3 - X_4$};
>
>   - If we simply remove $X_2$ and run PC on the remainder, we will obtain {$X_1 - X_3 - X_4$};
>
>   - If we simply remove $X_3$ and run PC on the remainder, we will obtain {$X_1 \rightarrow X_4 \leftarrow X_2$};
>
> The three resulting graphs above are not consistent with each other. This example illustrates that we can NOT simply remove determined variables so that the remainder contains no deterministic relations, then run normal causal discovery methods on the remainder, and then reintegrate the determined variables into the resulting graph as children of variables determining them -- there are many ways of removing determined variables, while the results on each may be neither consistent with each other, nor with the true graph.
>
> Second, regarding BMI, indeed, we agree with the reviewer that BMI can sometimes be regarded as an artifact/construct which is sufficiently calculable in terms of height and weight. On the other hand, BMI can also be seen as a variable with real causal meaning, as discussed in paper [59,60] -- BMI alone is involved in the causes of lung cancer. (Of course, one might argue that {BMI $\rightarrow V_{other}$} can be equivalently represented by {weight $\rightarrow V_{other} \leftarrow$ height}; However, that will result in a denser graph representation, violating the sparsity principle.)
>
> As noted above, we acknowledge that there is an ongoing debate about whether BMI is a variable or a construct. However, our focus here is not this specific example. **What we want to emphasize is that, usually in real-world datasets with deterministic relations, we CANNOT distinguish between the causal variables and constructs without prior knowledge.** For instance, here we tend to see BMI as a construct (and try to remove it) simply because we have the prior knowledge on its definition. However, without prior knowledge, simply from the data perspective, we cannot distinguish between weight, height, and BMI -- they hold equal status. If we choose to remove BMI, then why not remove weight or height (as they can also be calculated from the remaining two)? Removing either height or weight makes less sense (as we tend to understand them with "real causal meaning"), and will also make the resulting graph incorrect (e.g., there are many other variables only affected by height or weight); However, we cannot distinguish them from BMI. This scenario motivates the {X1,X2,X3,X4} example as discussed above.
>
> To summarize, we are neither trying to simply remove variables to eliminate deterministic relations (which may result in inconsistent results), nor trying to distinguish between causal variables and defined constructs (which is usually impossible without prior knowledge). Instead, we aim to put all variables together and recover the whole causal relations as well as the deterministic relations (up to equivalence class).
>
> ---
>
> [59] Eyal Shahar. "The association of body mass index with health outcomes: causal, inconsistent, or confounded?." American journal of epidemiology, 2009.
>
> [60] Robert Carreras-Torres, et al. "The causal relevance of body mass index in different histological types of lung cancer: a Mendelian randomization study." Scientific Reports, 2016.

---

> > ### Comment · Reviewer_zDFG · 2024-08-12
> >
> > Thank you for your response.
> >
> > You say that the three graphs are not consistent with each other, but I don't understand why this matters? Removing $X_3$ leaves us with a perfectly good fully identified graph with both fewer variables and fewer edges, making it *more sparse*. Similarly, including BMI leads to a more sparse causal graph only if it is a mediator for the causal effects of height and weight on multiple other variables.
> >
> > I understand that we can't tell a priori which variable to exclude, but my question is rather why we shouldn't do so after we have run your method

---

> ### Author Response · Authors · 2024-08-12
>
> Thank you so much for your valuable time and follow-up questions. Below are our responses.
>
> ```
> “Removing X3 leaves us with a perfectly good fully identified graph.”
> ```
>
> - We completely agree with you that removing $X_3$ would yield a well-identified graph for describing the joint distribution of $\{X_1, X_2, X_4\}$.
> - However, for causal discovery, we have to finally present a graph with all four variables (i.e., to add $X_3$ back into the graph) because, as mentioned above, we don't know a priori whether $X_3$ is a construct or a causal variable. Any variable may exactly be of the user's interest. We cannot delete any variable from the final result graph.
> - Then, how can we add $X_3$ back into the graph? Based on the result $\{X_1 \rightarrow X_4, X_2\rightarrow X_4 \}$, one intuitive way is to reintegrate $X_3$ back as a child of the variables that determine it, leading to a graph with the edges $\{X_1 \rightarrow X_4, X_2\rightarrow X_4, X_1\rightarrow X_3, X_2\rightarrow X_3 \}$. However, this results in a denser graph than the true underlying structure (that's how we justify "sparsity" above). Even worse, the critical information that $X_1$ and $X_2$ are affecting $X_4$ through $X_3$, is not shown in this reintegrated graph.
>
> To summarize, it is not recommended to remove and reintegrate a variable for causal discovery to deal with deterministic relations. Any variable could be of interest to the users, and we aim to discover causal relations among all variables.
>
> ```
> "Why shouldn’t we exclude determined variables after running your method?"
> ```
> - We totally agree that after running our method, one can exclude any determined variables as a postprocessing step. That is totally fine.
> - The core argument in our paper is that one cannot bypass our method by directly excluding determined variables and then applying standard causal discovery methods like PC on the remaining variables. The reasons are discussed above.

---

> > ### Comment · Reviewer_zDFG · 2024-08-13
> >
> > Thank you for your responses. I've updated my score.

---

> > > ### Author Response · Authors · 2024-08-13
> > > **Thank you very much for your valuable time and insightful comments**
> > >
> > > We sincerely appreciate the reviewer for carefully checking our responses and engaging in a fruitful discussion with us. Your constructive feedback has significantly improved our paper! Thanks a lot!

---

### Author Response · Authors · 2024-08-07
**Results on A New Real-world Dataset**

Dear Reviewers:

Thanks a lot for your thoughtful and constructive reviews. We are encouraged by some of your positive comments and inspired by your insightful suggestions. Here, we provide more details about our experiments on the new real-world dataset.

**Description:** We use US census Public Use Microdata Sample (PUMS), and follow the data preprocessing procedure outlined in [59], which is a modern version of the UCI Adult data set [60]. Datasets based on census data are widely considered in the algorithmic fairness literature [61, 62]. Here we choose 5 important variables, i.e., **{Age, Occupation, Sex, Annual_income (AI), and Adjusted_annual_income (AAI)}**, in total there are 3000 samples. Because of the potentially different timeframe of the survey cycle, AAI (= AI * Adjusted_factor) are the adjusted dollar amounts that they have earned entirely during the calendar year. Within one calendar year, this adjusted_factor is a constant. Here, we choose the data in 2021. Therefore, AAI and AI have a deterministic relation.

**Results:** We compare our DGES with GES and PC.

- The result of DGES is: {Sex $\rightarrow$ Occupation $\leftarrow$ AI, AI $\leftarrow$ AAI, Sex $\rightarrow$ AAI $\leftarrow$ Age}, 5 edges.

- The result of GES is:  {Sex $\leftarrow$ Occupation -- AAI, AI -- AAI -- Age, AI $\rightarrow$ Sex $\leftarrow$ AAI}, 6 edges.

-  The result of PC is: {Sex $\rightarrow$ Occupation $\leftarrow$ Age, AI -- AAI}, 3 edges.

**Analysis:** Compared with GES, the result of DGES is more sparse. Particularly, we can detect that AI and AAI have a deterministic relation, and GES gives redundant edges by {AI $\rightarrow$ Sex $\leftarrow$ AAI} while our DGES only keeps one edge {Sex $\rightarrow$ AAI}. Moreover, the result of PC is totally different from the other two. Clearly, in our DGES result, AI and AAI are still connected, and we can still see the BS, i.e., {AI $\rightarrow$ Occupation, AAI -- Sex, AAI--Age}. However, as a result of PC with FisherZ test, the BS becomes empty, which is exactly due to the violation of faithfulness.

Thanks again for the time you dedicated to carefully reviewing this paper. We hope our new experimental results can adequately address your concerns.


Sincerely, Authors of Submission 9142.

---
References:

[59] Frances Ding, et al. "Retiring adult: New datasets for fair machine learning." NeurIPS, 2021.

[60] Barry Becker and Ronny Kohavi. Adult. UCI Machine Learning Repository, 1996.

[61] Razieh Nabi and Ilya Shpitser. "Fair inference on outcomes." AAAI, 2018.

[62] Zeyu Tang, et al. "Procedural Fairness Through Decoupling Objectionable Data Generating Components." ICLR, 2024.

---

### Decision · Program_Chairs · 2024-09-25

**Decision:**

Accept (poster)

**Comment:**

In this manuscript, the authors study the problem of causal discovery in the presence of deterministic dependencies. The initial reviews were already quite positive, while during the rebuttal the authors convincingly addressed many questions and key concerns, leading to improved scores. The reviewers praise the clarity of the presentation, the relevance and novelty of the topic, as well as the technical treatment. Not all are equally impressed by the empirical evaluation, and I agree this could be improved upon both in terms of competing methods and data settings. All in all, this paper clears the acceptance threshold with a wide margin. I recommend the authors to use the additional page to address the questions that arose during the discussion and include additional results (such as provide presented during the discussion).